

Statistical Analysis of Contrail to Cirrus Evolution during the Contrail and Cirrus Experiments
(CONCERT)
Aurélien Chauvigné[1], Olivier Jourdan[1], Alfons Schwarzenboeck[1], Christophe Gourbeyre[1],
Christiane Voigt[2,3], Hans Schlager[2], Stefan Kaufmann[2], Stephan Borrmann[3,4], Sergej Molleker[3,4],
Andreas Minikin[2,5], Tina Jurkat[2], Ulrich Schumann[2]
[1]Laboratoire de Météorologie Physique, UMR 6016 CNRS/Université Clermont Auvergne,
Clermont-Ferrand, France.
[2]Institut für Physik der Atmosphäre, Deutsches Zentrum für Luft- und Raumfahrt (DLR),
Oberpfaffenhofen, Germany.
[3]Institut für Physik der Atmosphäre, Universität Mainz, Mainz, Germany.
[4]Max-Planck-Institute for Chemistry, Department for Particle Chemistry, Mainz, Germany.
[5]Now at: Flugexperimente, Deutsches Zentrum für Luft- und Raumfahrt (DLR), Oberpfaffenhofen,
Germany.
**Abstract:**
Air traffic affects the cloudiness, and thus the climate, by emitting exhaust gases and particles. The
study of the evolution of contrail properties is very challenging due to the complex interplay of vortex
dynamics and atmospheric environment (e.g. temperature, supersaturation). Despite substantial
progress in recent years, the optical, microphysical, and macrophysical properties of contrails and
ambient cirrus during contrail formation and subsequent ageing are still subject to large uncertainties
due to instrumental and observational limitations and the large number of variables influencing the
contrail life cycle. In this study, various contrail cases corresponding to different aircraft types and
atmospheric conditions are investigated using a statistical method based on the in situ optical
measurements performed during the CONCERT campaigns 2008 and 2011. These two aircraft
campaigns encompass more than 17 aircraft contrail cases. A Principal Component Analysis (PCA)
of the angular scattering coefficients measured by the Polar Nephelometer has been implemented in
order to classify the sampled ice cloud measurements in 6 clusters representative of different
development stages of the contrails (primary wake, young contrail, contrail-cirrus and natural cirrus).
Based on the information derived from air traffic control, extinction coefficients, asymmetry
coefficients, nitrogen oxide concentrations, relative humidity with respect to ice (RHI) and particle
size distributions are analyzed for each cluster to provide a characterization of the evolution of ice-
cloud properties during the contrail to cirrus evolution. The PCA demonstrates that contrail optical
properties are well suited to identify and discriminate the different contrail growth stages and to
provide an independent method for the characterization of the evolution of contrail properties.

# 1 Introduction

Aircraft exhaust plumes have a significant impact on climate and tropospheric chemistry (Lee et al.,
2010; IPCC, 1999). The Intergovernmental Panel for Climate Change IPCC special report on aviation
(1999) estimates that $NO_x$ emissions from subsonic aircraft increase ozone concentrations by up to
6% at cruise level. Short and long lived pollution species have different impact on atmospheric
chemical composition depending on the flight level (Frömming et al, 2012). Emissions of water
vapor, black carbon (BC) / soot particles, sulfate ($SO_4$) aerosols and nitrogen oxides ($NO_x$) contribute



to the modification of the chemical composition of the upper troposphere on shorter timescales (Lee
et al., 2010, Gettelman and Chen, 2013; Liou et al., 2013). The long term climate impact is mainly
driven by $CO_2$ emissions. Modelling studies have shown that the direct radiative forcing from aviation
is expected to represent 3-4% (50-60 mW m$^{-2}$) of the anthropogenic forcing (Lee et al., 2010; De
Leon et al., 2012) and could reach 87 mW m$^{-2}$ in 2025 (Chen and Gettelman, 2016). Aircraft induced
cloudiness has also an important impact on climate, although the quantitative assessment of the
radiative forcing remains a major source of uncertainties (Lee et al., 2010).
Contrail formation is mainly controlled by the thermodynamic properties of the ambient air and by
the aircraft emissions. The conditions for contrail formation can be determined by the Schmidt-
Appleman Criterion (SAC) (Schumann, 1996). Contrail chemical composition can have a significant
impact on the contrail formation (Kärcher et al., 2009). Indeed, the emission index (e.g. soot emission
index in kg-fuel$^{-1}$) is directly linked to the contrail microphysical properties, as the total number
densities and ice crystal diameters. Several studies in the past have been dedicated to the evolution of
concentrations of nitrogen oxide (NO) and sulfur dioxide ($SO_2$). Rapidly oxidized by OH (post-
combustor reaction; Jurkat et al., 2011), NO and $SO_2$ are transformed into a series of species including
nitrous acid (HONO), nitric acid ($HNO_3$), and sulfuric acid ($H_2SO_4$). Part of the nitric acid, e.g.,
formed in the young plume is taken up by contrail ice particles (Kärcher and Voigt, 2006; Voigt et
al., 2006; Schäuble et al., 2009). HONO can be a temporary reservoir of OH by photolysis reactions,
and $H_2SO_4$ a precursor for radiatively active sulfate particles also contributing to soot particle coating
(Jurkat et al., 2011). OH-induced reactions, formation and reaction schemes of greenhouse gases
($H_2O$, $O_3$, $CH_4$), as well as emission of black carbon and sulfate aerosols have significant impact on
climate (Frömming et al., 2012; Gettelmann and Chen, 2013). Two different processes of contrail
formation have been studied: combustion condensation trails and aerodynamic condensation trails.
Different studies (Gierens and Dilger, 2013; Jansen and Heymsfield, 2015) have illustrated
characteristics of aerodynamically controlled contrail formation associated to warmer temperatures
(observations at temperatures above -38°C). For contrails primarily initiated by the combustion
processes, the mixing of hot and humid exhaust gases with cooler and dryer ambient air increases the
local relative humidity in the exhaust plume leading to the formation of contrails when the saturation
with respect to liquid water is reached. Thus, soot and sulfate aerosols emitted by the aircraft (Moore
et al., 2017) may act as condensation nuclei to form liquid droplets. Homogeneous ice nucleation of
the liquid droplets can occur when the exhaust cools down through mixing with the ambient
temperature, while preserving ice saturation. Small ice crystals are then formed in the jet phase within
some tenths of a second (Kaercher and Yu, 2009). The further life-cycle of contrails depends on the
interaction with the wake vortices behind aircraft and the ambient atmosphere (Irvine et al., 2012;
Graf et al., 2012; Duda et al., 2013; Carleton et al., 2013; Schumann and Heymsfield, 2017). The ice
crystals in the young contrails  are captured within two counter-rotating wake vortices in the
downwash behind the aircraft induced by the aircraft lift, which induce adiabatic compression,
heating, and partial sublimation of the ice crystals within the primary wake (Lewellen and Lewellen,
2001; Sussmann and Gierens, 2001, Unterstrasser et al., 2008, Unterstrasser et al., 2016; Kärcher and
Voigt, 2017). This primary wake may disappear if ambient air is subsaturated with respect to ice. In
the case of supersaturation, the secondary wake becomes visible, thereby detraining ice particles from
the primary wake at a higher level (Sussmann and Gierens, 1999, Kaufmann et al., 2014). The initially
almost spherical ice crystals become increasingly aspherical and grow by uptake of water vapor as
long as saturation with respect to ice is prevailing. For ambient humidity above ice saturation,
contrails can persist after the vortex breakdown, spread and evolve into contrail cirrus (Schumann
and Heymsfield, 2017). The associated cloud cover (larger than for linear contrails alone) of contrail
cirrus then shows increasing impact on the radiative forcing (Burkhardt and Kärcher, 2011;
Schumann et al., 2015).



The assessment of the contrail radiative forcing requires, in particular, an accurate estimation of the
cloud cover, the visible optical depth, the single scattering characteristics, the ice crystal effective
size and habit (Yang et al., 2010; Spangenberg et al., 2013). Satellite observations provide a
comprehensive dataset to study statistically the contrail to cirrus evolution. The combined contrail
tracking algorithms on the Spinning Enhanced Visible and Infrared Imager (SEVIRI) on board the
Meteosat Second Generation (MSG) satellites with properties inferred by the Moderate Imaging
Spectroradiometer (MODIS) on board the Terra satellite was used by Vazquez-Navarro et al., (2015)
to characterize the properties of 2300 contrails. Properties included lifetime (mean values of 1h), the
width (8 km), the length (130 km), the optical thickness (0.34), the altitude (11.7 km) and the radiative
forcing (-26 W m$^{-2}$ for shortwave forcing over land) of these contrails. However, detailed in situ
optical and microphysical measurements are still needed to evaluate satellite products and to develop
more appropriate retrieval algorithm. In particular, distinguishing contrails from natural cirrus
remains extremely challenging from satellite observations. Although the optical and microphysical
properties of young contrails (linear contrails) differ from natural cirrus properties, the contrail
properties are highly time dependent and persistent contrail cirrus can be embedded in thin cirrus
clouds. Recent *in situ* measurements (Voigt et al., 2017) show that the microphysical properties of
contrail cirrus can still be distinguished from natural cirrus at contrail cirrus ages up to several hours.
Most of the studies (Jessberger et al., 2013; Lewellen et al., 2012; Schumann et al., 2013) separate
the contrail analysis between the two wakes. Indeed, the primary and secondary wake properties
depend strongly on atmospheric conditions and aircraft type (emission index, vortex, flight level,
ambient humidity, temperature,…). In the primary wake, contrail ice crystals are quasi-spherical with
values of the effective diameter (Deff) typically lower than 4 µm (Schumann et al., 2011; Gayet et
al., 2012; Järvinen et al., 2016; Schumann et al., 2017b). Also the total number concentration of ice
particles is typically larger than 1000 cm$^{-3}$ a few seconds after contrail formation (Baumgardner and
Gandrud, 1998; Petzold et al., 1997) and subsequently decreases by dilution to concentrations below
200 cm$^{-3}$ within less than a minute after contrail generation (Poellot et al., 1999; Schröder et al., 2000;
Gayet et al., 2012). Gayet et al. (2012) reported mean values of ice water content of 3 mg m$^{-3}$ and
maximum extinction coefficients close to 7 km$^{-1}$. The recent overview on contrail studies presented
in Schumann et al. (2017b) reports several microphysical properties at different stages, for different
atmospheric conditions as well as comparisons with the Contrail Cirrus Prediction (CoCIP) model
simulations. Their study highlights a large variability (which increases with contrail age) of contrail
properties. Comparing primary and secondary wakes, several studies reported findings on the
secondary wake and its evolution into contrail cirrus. Detrained from the primary wake and submitted
to saturated ambient air with respect to ice, ice crystals grow rapidly, while crystal concentration
decreases. Within the first minutes after formation, measurements exhibit aspherical ice crystals
characterized by effective sizes up to 6 µm, IWC ranging between 2.5 and 10 mg m$^{-3}$, extinction
between 2 and 3 km$^{-1}$, with crystal concentrations typically lower than 100 cm$^{-3}$ (Goodman et al.,
1998; Voigt et al., 2010; Kübbeler et al., 2011; Gayet et al., 2012; Jeßberger et al., 2013; Schumann
et al., 2013; Poellot et al., 1999; Febvre et al., 2009; Kaufmann et al., 2014). Aged contrails can persist
and evolve into contrail cirrus if the ambient air is saturated with respect to ice, however those studies
are limited by the lack of unambiguous identification (Schumann et al., 2017a). Also after a few
minutes, difficulties appear for the pilot to track the contrail by visual navigation, which is due to
contrail and contrail cirrus spreading in the free troposphere. Observations of the ice crystal shape
and growth over several tens of minutes and up to an hour illustrate that crystal effective size can
easily reach 20 µm and beyond with number concentrations ranging from 1 to 5 cm$^{-3}$ (Lawson et al.,
1998; Schäuble et al., 2009), extinction less than 0.5 km$^{-1}$ (Febvre et al., 2009), and IWC up to 10
mg m$^{-3}$ (Schröder et al., 2000; De Leon et al., 2012). At this stage, within a sustained ice-
supersaturated environment, contrail microphysical properties may still differ from those of natural
cirrus (Voigt et al., 2017) with concentrations of ice crystals larger than 100 µm in the order of 0.1



cm$^{-3}$. These crystals typically show bullet rosette type habits (Heymsfield et al., 1998; Heymsfield et
al., 2010). Optical depth values can reach a value of 2.3 (Atlas and Wang, 2010), corresponding to an
extinction of 0.023 km$^{-1}$. Nevertheless, the transition from contrails to cirrus highly depends on the
ambient saturation conditions and modelling studies with typical atmospheric conditions suggest time
evolution of optical and microphysical properties from contrail to contrail cirrus clouds (Burkhardt
and Kärcher, 2011; Unterstrasser et al., 2016 ; Schumann et al., 2015).
In this study, we report on a method presenting a powerful alternative for classifying cloud events
into young contrail, contrail-cirrus and natural cirrus. The method is applied to aircraft data of the
CONCERT (Contrail and Cirrus Experiment) campaigns (Voigt et al., 2010, 2011, 2014). The
methodology consists of implementing a principal component analysis (PCA) of the angular light
scattering data from a Polar Nephelometer. The PCA results of the different type of contrails (different
clusters) are then utilized with corresponding optical, microphysical, and chemical properties in order
to validate hypothesis on contrail phase definitions (young contrails to cirrus contrails). This paper
starts with an illustration of the properties of contrails and cirrus clouds observed during two specific
CONCERT flights (19 November 2008 and 16 September 2011) encompassing a series of different
contrail evolution phases. These two flights containing a variety of contrail-cirrus information can be
regarded as an analytical framework producing results which then can be compared to contrail-cirrus
properties of other flights.

## 159 2 CONCERT projects and data processing

### 160 2.1 CONCERT campaigns

CONCERT-1 and CONCERT-2 campaigns took place in October/November 2008 and
August/September 2011, respectively. These two campaigns with the DLR Falcon 20 E research
aircraft were based in Oberpfaffenhofen, Germany, and sampled contrails and cirrus at mid-latitudes
in the Northern Hemisphere. The overall objective has been to reduce uncertainties on the
microphysical, chemical, and radiative properties of contrails behind aircraft of different types and to
improve the evaluation of contrail's impact on climate. Besides the primary objectives focusing on
contrails, few CONCERT flights were dedicated to emissions of Etna and Stromboli volcanos (Voigt
et al., 2014; Shcherbakov et al., 2016). Also a few stratospheric intrusions were observed during the
flight missions. In total, 23 flights were recorded during the two measurement campaigns, wherein
12 flights were entirely focused on aircraft contrail chasing. Overall, more than 17 different aircraft
exhausts plumes have been probed. Particularly, the CONCERT-2 campaign mainly focused on
observing persistent contrails, and hence on the evolution of contrails into contrail cirrus.
During both CONCERT campaigns, the DLR research aircraft Falcon was equipped with a set of
instruments to measure the optical and microphysical properties of cloud particles and also the trace
gas composition in the UTLS (Upper Troposphere / Lower Stratosphere) region. Voigt et al. (2010)
provide a detailed description of the aircraft instrumentation. We briefly introduce the instruments
used in this study.

### 178 2.2 Aircraft instrumentation

The microphysical and optical particle properties of contrails and cirrus presented in this study were
mainly derived from the PMS Forward Scattering Spectrometer Probe 300 (FSSP-300), the Polar
Nephelometer (PN), and the PMS 2D-C hydrometeor imaging probe. The combination of these
independent techniques characterizes cloud particles within a range of diameters varying from 0.5
μm to 2 mm.



The PN (Gayet et al., 1997) measures the angular scattering coefficients (non-normalized scattering
phase function) of an ensemble of water droplets or ice crystals or a mixture of those particles ranging
from a few micrometers to approximately 1 mm in diameter. These particles intersect a collimated
laser beam, at a wavelength of 804 nm, near the focal point of a parabolic mirror. The light scattered
at angles from 3.49° to 172.5° is reflected onto a circular array of 56 near-uniformly positioned
photodiodes. In this study, reliable measurements were performed at 30 scattering angles ranging
from ±15° to ±162°. The measurements allow to distinguish particle phase (water droplets or ice
crystals) and to derive single scattering properties such as the extinction coefficient and the
asymmetry coefficient with uncertainties of 25% and 4%, respectively (Gayet et al., 2002; Jourdan et
al., 2010).
Particle size distributions and corresponding microphysical and optical integrated properties (IWC,
Deff, N, and extinction) were derived from FSSP-300 measurements (Baumgardner et al., 1992). This
instrument measures the intensity of forward scattered light from cloud particles passing through the
laser beam, with cloud particles in the diameter range 0.35-20 µm. In the forward angular region
(from 4° to 12°), scattering is mainly described by the particle diffraction pattern and therefore
depends on the refractive index, the shape, and the size of the particles. The method of data processing
and size calibration used during the CONCERT campaigns have been presented in Gayet et al. (2012)
(Appendix A). We briefly recall that the asymmetry parameter derived from the PN was used to
discriminate nearly spherical particles (g ≥ 0.85) from non-spherical ones (g < 0.85) at 804 nm. For
spherical ice particles, Mie calculations were used to derive the size bin limits and the corresponding
extinction efficiency. Results were adjusted to the calibrated probe response. Additionally, in order
to minimize Mie ambiguities related to the FSSP-300 size response, 31 channels were rebinned to 13
channels with a diameter ranging from 0.5 µm to 18 µm (upper channels 30 and 31 were excluded
from the data analysis). For non-spherical particles, the size of the contrail particles is expressed in
terms of an equivalent surface or area diameter, i.e. the diameter of a sphere that has the same area
than the projected area of the measured non spherical particle image (Mishchencko et al., 1997;
Schumann et al., 2011). The particles were assumed to be rotationally symmetric ice ellipsoids with
an aspect ratio of 0.5. Accordingly, and contrary to the method used for spherical particles, 15 size
bins ranging from 0.5 µm to 18 µm were defined based on T-Matrix calculations following Borrmann
et al., (2000).
The bi-dimensional optical array spectrometer probe (2DC) provides information on the crystal size
and shape within a nominal size range from 25 µm to 800 µm by recording cloud particles shadow
images with a 25 µm resolution. The method of data processing used in this study is described in
detail in Gayet et al. (1996) and Febvre et al. (2009). Reconstruction of truncated particles has been
considered for the PSD calculations and the sampling surfaces have been derived according to
Heymsfield and Parrish (1978). In order to improve the statistical significance of low particle
concentrations, a 5-s running mean was applied. The bulk parameters were calculated assuming the
surface-equivalent diameter relationships of Heymsfield (1972) and Locatelli and Hobbs, (1974). As
the sensitivity of the probe to small particles decreases with airspeed (Lawson et al., 2006), particles
smaller than 100 µm may not be detectable at the Falcon airspeed of typically 180 m s$^{-1}$. This may
result in larger uncertainties of up to 100% in the derived microphysical parameters such as the IWC
(Gayet et al., 2002 and 2004).
Depending on the spherical or non-spherical shape of ice crystals, ice water content IWC, extinction
coefficient Ext, and effective diameter $D_{eff}$ were calculated independently according to Garret et al.
(2003) and Gayet et al. (2012). For spherical ice crystals (gPN ≥ 0.85), optical and microphysical
properties are calculated from the following equations:



$$Ext = \frac{\pi}{4} \sum_i \beta_{ext}^i N_i D_i^2 \tag{1}$$

$$IWC_{spherical} = \frac{\pi}{6} \rho_{ice} \sum_i N_i D_i^3 \tag{2}$$

where $\beta_{ext}^i$ is the extinction efficiency (depending on spherical or aspherical particle characterization), $D_i$ the mean diameter in channel i, $N_i$ the number concentration, and $\rho_{ice}$ the bulk ice density (0.917 g cm$^{-3}$).

For non-spherical ice crystals (gPN < 0.85 and for particle diameters larger than 70 µm), an equivalent diameter method is used (Gayet et al., 2004). For an ice crystal with an area A, the particle equivalent diameter $D_{equ}$ and equivalent mass $x_{equ}$ are defined as :

$$A \leq 0.049 \text{ mm}^2 \qquad\qquad D_{equ} = 0.82\, A^{0.48} \tag{3}$$

$$A > 0.049 \text{ mm}^2 \qquad\qquad D_{equ} = 0.56\, A^{0.32} \tag{4}$$

$$x_{equ} = \frac{\pi}{6} \rho_{water} D_{equ}{}^3 \tag{5}$$

$$IWC_{non-spherical} = \rho_{ice} \sum_i N_i x_{equ} \rho_{water} \tag{6}$$

With $\rho_{water}$ the bulk water density (1 g cm$^{-3}$).

In young tropospheric aircraft plumes, the main chemical component to be measured is $NO_y$, mainly composed of NO and $NO_2$. During CONCERT campaigns trace gas measurements of NO and $NO_y$ mixing ratio were performed using the chemiluminescence technique (Schlager et al., 1997) with a time resolution of 1 s. The accuracy (and precision) of the NO and $NO_y$ measurements are estimated with 7% (and 10%) and 10% (and 15%), respectively (Ziereis et al., 2000).

Chemical ionization mass spectrometry combined with a $SF_5^-$ ion source was used to detect the concentration of $HNO_3$, $SO_2$, and HONO in the exhaust plumes and the UTLS (Jurkat et al., 2010, Jurkat et al., 2011). Mass spectra were sampled with an ion trap mass spectrometer with resolution of < 0.3 atomic mass units and averaged over five spectra resulting in an overall time resolution of 1.6 s. Detection limits for HONO and $SO_2$ for 1.6 s time resolution were 72 and 67 pmol mol$^{-1}$ and for $HNO_3$ 36 pmol mol$^{-1}$ for 32 s time resolution. During CONCERT 2011 a quadrupole mass spectrometer was employed on the Falcon (Voigt et al., 2014) with detection limits of 15 pmol mol$^{-1}$ for $HNO_3$ and 8 pmol mol$^{-1}$ for $SO_2$ at 20 s time resolution (Jurkat et al., 2016).

The detection of water vapor and relative humidity with respect to ice (RHI) is important to characterize contrail ice crystals. Water vapor has been measured with the chemical ionization mass spectrometer AIMS-H2O during CONCERT-2 (Kaufmann et al., 2014; 2016). In addition, hygrometers using the Lyman-α technique (Zöger et al., 1999; Meyer et al., 2015), and frost point hygrometers (Heller et al., 2017) were implemented on the Falcon during CONCERT-1 and 2.

## 3   Results

### 3.1 Overview of the cloud properties sampled during the reference cases





The purpose of this section is to give an overview of the contrail optical properties and more
interestingly to evaluate the ability of the Polar Nephelometer measurements to identify contrails.
Two flights, performed on 16 September 2011 during CONCERT-2 (flight 16b) and on 19 November
2008 during CONCERT-1 (flight 19b), respectively, were selected for their variety of contrails and
cirrus sampled during these two flights. The two flights are considered as a benchmark to illustrate
the potential of the PCA methodology described in Sect. 3.2.

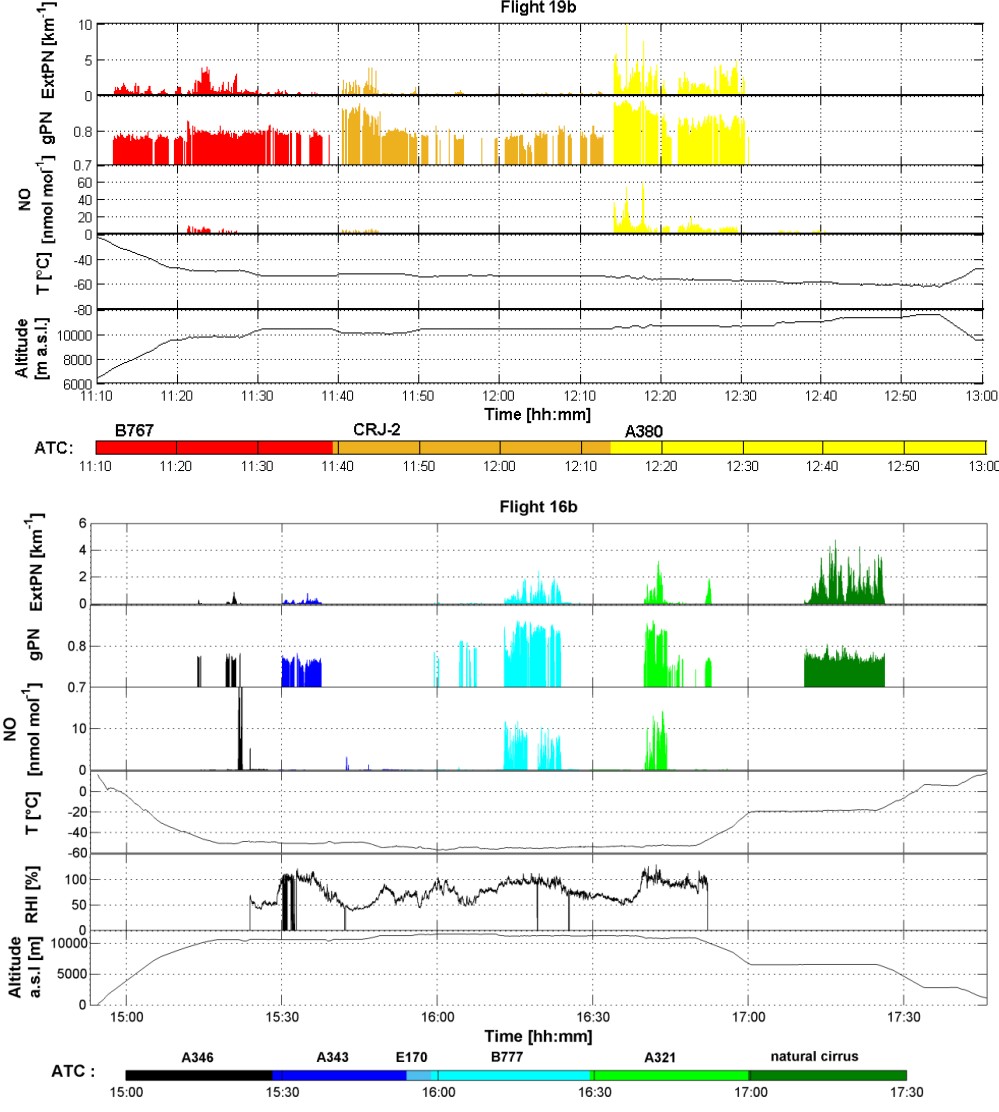

Figure 1 : Time series at 1 s resolution for flights a) 19b and b) 16b. From top to bottom: extinction coefficient (in km$^{-1}$) and asymmetry parameter measured by the Polar Nephelometer at 804 nm, concentration of nitric oxide (in nmol mol$^{-1}$) measured by chemiluminescence technique, temperature (in °C), relative humidity with respect to ice (in %), and altitude a.s.l. (in m). Temporal series are colored according to time, and grey indicators illustrate contrail information from Air Traffic Control (ATC) information as provided by a dedicated flight controller.





Figure 1 displays the time series of the extinction coefficient (ExtPN) at 804 nm, asymmetry
parameter (gPN) at 804 nm, relative humidity (RHI), and the nitric oxide (NO) concentration for
flights 19b and 16b. RHI measured with the AIMS mass spectrometer is shown for flight 16b. For
RHI measurements on flight 19b we refer to Kübbeler et al. (2011); Gayet et al. (2012); Jessberger et
al. (2013) ; Schumann et al. (2013). For both flights, Air Traffic Control (ATC) provided information
on the aircraft characteristics (aircraft type, engine type, fuel flow, weight, engine power setting,…)
responsible for the formation of encountered contrails. In a first classification approach, the time
series are color coded according to the ATC information of consecutive sampling of different
contrails.
The PN Extinction coefficient coupled with the asymmetry parameter seems to be a reasonable proxy
to detect contrails and cirrus clouds (see amongst other references, Voigt et al., 2010). ExtPN values,
by definition, depend on the cloud particle concentration and size. Values typically beyond $0.1\ \text{km}^{-1}$
correspond in general to cloud events well correlated to supersaturation with respect to ice conditions
(RHI > 100%).  Figure 1 proves that relatively high values of extinction can be found in flights 19b
and 16b which are linked to the presence of contrails or cirrus clouds. Moreover, the temporal
distributions of these values are coherent with ATC information for both flights. For instance, most
of the contrails induced by commercial aircraft exhaust plumes translate into significant extinction
coefficient values. The ExtPN values are between $0.2\ \text{km}^{-1}$ and $10\ \text{km}^{-1}$ for contrails induced by A346,
A340, and A380 commercial aircraft. Cirrus clouds are detected with more variable extinction values
mostly larger than $0.5\ \text{km}^{-1}$. Most of the aircraft induced contrails are detected by the PN with  the
exception of the ones stemming from the E170 airplane. At 15:50 during flight 16b, ATC identified
the E170 position close to the Falcon flight trajectory, however the ExtPN and the NO mixing ratio
remained very low. Hence,  the E170 contrail was not probed by Falcon. In the following we assume
that only periods with ExtPN values above $0.1\ \text{km}^{-1}$ are considered a reasonably reliable signature for
contrails sampled during the flight campaigns.
The absolute values of the asymmetry parameter gPN provide additional information of the cloud
particle shape. Indeed, gPN is a good indicator of the degree of sphericity of ice crystals (Gayet et al.,
2012). Ice clouds with gPN values higher or equal to 0.85 are typically composed of spherical ice
crystals, whereas lower values are indicative of aspherical ice particles. In a supersaturated
environment of contrails, crystals grow by water vapor deposition and become increasingly aspherical
with time. This is why spherical ice crystals prevail in very young contrails with an asymmetry
coefficient around 0.85 with RHI above 100%. Subsequently, gPN is decreasing when water vapor
diffusion is generating more and more aspherical crystal shapes at ice supersaturation. This can be
observed for A321 chasing during flight 16b with gPN decreasing to 0.75 whilst RHI remains around
100%, whereas for B777 chasing, no gPN decrease is observed at RHI < 80%. Also natural cirrus
clouds are mainly composed of non-spherical ice crystals, possibly with hexagonal shapes. These
clouds can be easily discriminated from young contrails as they exhibit a much lower asymmetry
parameter typically below 0.75 (see amongst others Jourdan et al., 2003b, Febvre et al., 2009).
However, no accurate ambient RHI data can retrieved for measurements in natural cirrus due to
instrumental calibration problems. A good example of the evolution of gPN is the CRJ-2 contrail
observed between 11:40 and 11:45 during flight 19b. The sequence illustrates the potential of the
gPN measurement to characterize the evolution of contrail properties, with decreasing crystal
sphericity documented by the decreasing asymmetry parameter from 0.88 to 0.79 (uncertainties
around 0.04) after only 5 min and down to 0.77 after 20 min. A more stable variation of gPN values
(around $0.78 \pm 0.02$) is then observed until 12:10 after 30 min of contrail ageing associated with
crystal growth by water vapor diffusion. A similar decrease in gPN has been noted by Gayet et al.
(2012) in the ageing contrail from an A380 aircraft and is visible for the B767 and the A321 contrails.





NO concentration measurements can also be used to discriminate natural cirrus clouds from ice clouds
influenced by aircraft traffic. At the typical altitude of 10 km, NO environmental concentrations are
close to background values. In contrast, NO concentrations in young contrails may reach several tens
of nmol mol$^{-1}$ (Voigt et al., 2010). Figure 1 shows a good correlation between the expected
localization of young contrails and NO concentrations. The dilution effect into the upper troposphere
causes an important decay of chemical concentrations. For instance, the first few seconds of the A380
chasing during flight 19b are characterized by a high NO concentration (up to 40 nmol mol$^{-1}$) followed
by a fast decrease to 10 nmol mol$^{-1}$ in the next 15 min, and less than 5 nmol mol$^{-1}$ beyond 15 min.
NO concentrations finally decrease to background levels within hours (e.g. Voigt et al., 2017). This
decrease of the NO concentration is in accordance with the decrease of the extinction coefficient
(from 10 to 0.2 km$^{-1}$) and asymmetry parameter (from 0.88 to 0.77). Thus NO was mainly used as
additional contrail indicator. However, during some aircraft chasing events, NO concentrations were
near background levels, while mass spectrometric measurements (not shown here) indicate elevated
concentrations of HONO, HNO$_3$, and SO$_2$ representative for contrail chemical species.
The above case studies of flights 19b and 16b clearly show that the optical properties of contrail type
ice clouds (supported by the ATC information) in conjunction with specific trace gas concentration
measurements can be used to discriminate contrails from natural ice cloud events. A first order
analysis of these parameters can be used to roughly distinguish young contrails (mostly quasi-
spherical ice crystals) from aged contrails (mostly aspherical ice crystals) and natural cirrus
(background NO concentrations). This analysis is mainly qualitative and based solely on a few
integrated parameters (Fig. 1). A more robust statistical method should be used to accurately separate
the different contrail phases and also natural cirrus. In the following section, relationships between
contrail and ice cloud properties and their scattering properties are investigated more extensively to
assess whether the information content of the PN scattering measurements is sufficient to document
changes in the contrail microphysical properties.
**3.2 Statistical Method**
In this section, we present a methodology based on the statistical analysis of the optical signature of
ice clouds and in particular contrails. The goal is to classify the contrail properties according to the
aircraft origin and evolution stage. The main objective of the Principal Component Analysis (PCA)
is data reduction in order to allow a better physical interpretation of the light scattering patterns
derived from the Polar Nephelometer measurements (Legendre and Legendre, 1998; Jourdan et al.,
2003). In this study, optical properties of ice crystals in the evolving contrail environment are
analyzed to evaluate contrail evolution. This statistical analysis was already successfully applied to
discriminate mixed phase clouds (Jourdan et al., 2010), liquid clouds, and ice clouds, (Jourdan et al.,
2003) as well as to characterize porous aerosol in degassing plumes (Shcherbakov et al., 2016) using
light-scattering properties measured by the Polar Nephelometer.
**3.2.1 Reference definition**
The PCA is first applied to the PN angular scattering coefficients measurements performed during
flights 16b and 19b which are here considered as our reference datasets. Initially, a correlation matrix
is calculated to characterize the link between each scattering angle. The PCA is designed to generate
a new limited set of uncorrelated parameters, called principal components $C_{lj}$ representative of the
original data set variability.
A first implementation of the PCA is performed to detect unreliable data or out of order photodiodes.
For instance, seven photodiodes presented a low signal to noise ratio and were excluded from the




dataset. Flight sequences characterized by ExtPN<0.1 were also removed. Finally, flight sequences
dedicated to aircraft chasing and ice cloud sampling were considered to perform a second PCA.
Then, the analysis is performed on the remaining angular scattering coefficients (4669 Angular
Scattering Coefficients (ASC) representing PN measurements of flights 16b and 19b) now restricted
to 25 angles $\theta$ ranging from 15° to 155°. The new set of variables or coordinates, $C_{lj}$, can be expressed
with the scalar product of the vector of reduced angular scattering coefficients $\overrightarrow{\sigma_j}(\theta)$ for the $j^{th}$
measurements, expressed in log scale, and the $l^{th}$ eigenvector $\xi_l(\theta)$ (i.e. principal component) of the
total data set correlation matrix (Jourdan et al., 2010).

$$C_{lj} = (\overrightarrow{\ln \sigma_j} - \langle \overrightarrow{\ln \sigma} \rangle)^T . \overrightarrow{\xi_l} \qquad (4)$$

where $\langle \overrightarrow{\ln \sigma} \rangle$ represents the average ASC of the dataset.
The first three eigenvectors $\overrightarrow{\xi_l(\theta)}$ of the correlation matrix are displayed in Fig. 2 along with their
normalized eigenvalues $\lambda_l$, representing more than 99% of the variability of the PN angular scattering
coefficients (ASC).

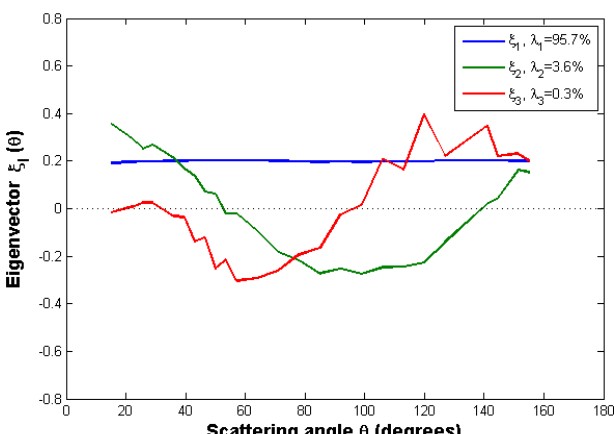

Figure 2 : First three eigenvectors for the flights 16b and 19b.

The first eigenvector $\xi_1(\theta)$ is approximately constant versus scattering angle and represents 95.7%
of the total variance. It means that this principal component is representative of changes of the
magnitude of phase functions without any changes in their global shape. This behavior means that
95.7% of the ASC variations are linked to changes of the cloud particle extinction. Results show a
good correlation ($r^2 = 0.98$) between the first eigenvector and the extinction derived from the PN
measurements (ExtPN).
The second eigenvector $\xi_2(\theta)$ reverses sign twice at scattering angles equal to 50° and 140° with an
extremum around 90°. Accordingly, 3.6% of the angular scattering variability corresponds to a
redistribution of scattered energy from the angular region (50°-140°) to scattering angles lower than
50° and higher than 140°. Light-scattering modeling studies demonstrate that the scattering behavior
in the angular region between 60° and 140° is sensitive to the particle shape and thermodynamic
phase (Jourdan et al., 2010). A strong linear correlation ($r^2=0.97$) between the second eigenvector and
the asymmetry coefficient (gPN) at 804 nm is found.





The third eigenvector represents only 0.3% of the total variance. However, this eigenvector provides
additional information in scattering regions which are not well described by the first two principal
components. It has opposite signs in the angular region (30°-90°) and (90°-155°) with maximum
extremal values at 60° and 120°. The shape of the third eigenvector describes the forward/backward
hemisphere partitioning of the scattering. Baran et al. (2012), Xie et al. (2006), and Xie et al. (2009)
showed that the scatter pattern for angles between 120° and 160°, corresponding to ice bow-like
effects, is sensitive to quasi-spherical particles. Moreover, these backscattering angles (θ>120°) and
scattering angles around 22° and 46° (corresponding to halo features) can also be linked to the particle
habits and surface roughness (Xie et al., 2009, Jourdan et al., 2010).
Based on these three first principal components, each phase function (or ASC) measured by the PN

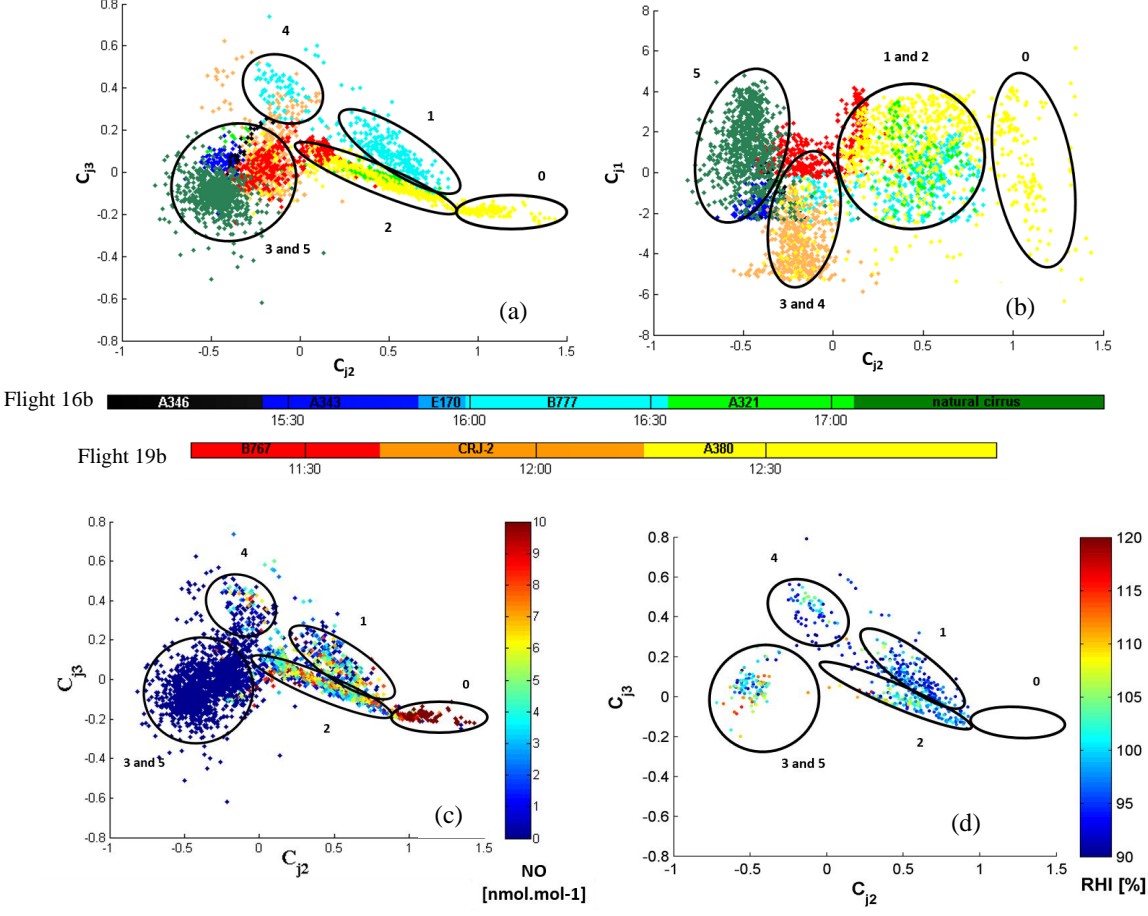

Figure 3: Expansion coefficient diagram for flights 16b and 19b (excluding 19b for RHI information): third versus second principal component for a), c) and d), and first versus second principal component for b). Data points are color coded according to ATC information for a) and b), by NO concentration for c), and by RHI values for d). The 6 typical scattering regimes (0-5) are indicated and numbered accordingly.

can be expressed with a good accuracy as a linear combination of these components (Jourdan et al.,
2010). The PN data are projected into a new space defined by the three principal components (3D-
space) instead of the 25-dimensional space of ASC. The scatterplots of the $C_{j3}$ and $C_{j1}$ expansion





coefficients versus the $C_{j2}$ coefficient are represented on Fig. 3a and b respectively. Fig. 3a illustrates
the features of the ASC measurements in one of the most comprehensive way. Each point corresponds
to a measured phase function documented over 25 angles. The variability of $C_{j2}$ coefficients is
significant with values ranging from -1 to 1.5. The angular variation of the second principal
component indicates that large values of $C_{j2}$ ($C_{j2} > 0.75$) correspond to ASC with low side scattering
(60°-130°) and higher forward scattering (15°-40°) and somehow higher backscattering (145-155°).
This behavior is connected to an increase of the asymmetry parameter with an increase of $C_{j2}$ values.
Thus, the fraction of spherical particles increases with increasing $C_{j2}$. In the region defined by
negative values of $C_{j2}$ the density of points is relatively high. These cloud events exhibit optical
properties characterized by a large side scattering and low asymmetry parameter. Therefore, specific
cloud sequences sharing similar scattering properties can be identified based on this second principal
component. Young contrails characterized by quasi-spherical ice crystals have high positive values
of $C_{j2}$ while cirrus clouds and contrail cirrus exhibit high negative values.
In the space of the third principal component high positive values of $C_{j3}$ imply that less energy is
scattering in the forward hemisphere and thus more energy is scattered in the backward hemisphere.
The variability of the expansion coefficients is less pronounced as ASC are distributed between -0.4
and 0.6. Most of the measured ASC do not significantly differ from the average ASC in the angular
ranges (30°-90°) and (90°-155°). However, some specific clusters linked to scattering behavior can
be identified for values of $C_{j3}$ greater than 0.1 and lower than -0.1. These threshold values also depend
of the position of the ASC on the second principal component. Finally, the first principal component
is directly linked to the extinction coefficient. High values of $C_{j1}$ are representative of optically dense
cloud sequences.
Based on the time series displayed in Fig. 1, data points corresponding to contrails are also color
coded according to their aircraft origin illustrated on Fig. 3a and b. From these information and based
on the first three principal components, 6 clusters (see numbered ellipsoids in Fig. 3) representative
of particular scattering behavior can be roughly identified. Figure 3a suggests an increase of $C_{j2}$ and
a decrease of $C_{j1}$ with increasing aircraft size. Figure 3c shows an increase of $C_{j2}$ for increasing NO
mixing ratio. Some contrails or ice cloud events are clearly delimited by a single area in the $C_{j3}$ versus
$C_{j2}$ and also $C_{j2}$ versus $C_{j1}$ diagrams. For instance cirrus clouds are gathered in cluster 5. Most of the
contrails induced by the B767, A340 and CRJ2 aircraft are associated to cluster 3 or 5. It means that
these cloud events share similar optical properties characterized by a low asymmetry parameter, high
side scattering behavior, and supersaturated ambient conditions with respect to ice. More
interestingly, Fig. 3 shows that some contrail events are smeared out over several areas or clusters.
Contrails relative to the A380 aircraft are dispatched in cluster 0 and 2 while the ones corresponding
to the B777 are spread out between clusters 1 and 4. This clearly indicates that the contrails are
evolving in space and/or time along the Falcon flight track while chasing the respective contrails.
This evolution can also be seen in the in-situ measurements of NO concentration color coded on Fig.
3c. Cloud regions influenced by air traffic can be discriminated from clouds formed by natural
processes based on the NO concentration values. While clusters 3 and 5 are characterized by very
low NO concentrations (close to zero) above background, clusters 0, 1, 2, and 4 correspond to higher
concentrations representative of a significant aircraft exhaust influence. For instance, a clear trend
shows that an increase of NO concentration translates into higher values of $C_{j2}$. Hence, contrails
characterized by a low side scattering due to the presence of spherical ice crystals correspond to high
NO concentration. This behavior can be a signature of young contrail properties. Elder or aged
contrails composed of a higher fraction of non-spherical crystals or growing more aspherically are
expected to exhibit an enhanced side scattering and a lower asymmetry parameter associated to lower
NO concentrations.



The contrail and cirrus classification based on ASC measurements appears to be consistent with the
independent trace gas measurements. Each cluster represented on Fig. 3 can be linked to a distinct
cloud event. Therefore, the combination of flights 16b and 19b can provide a relevant test-bed
database to discriminate contrail properties. Young contrails (spherical ice crystals) are associated to
clusters 0, 1 or 2, whereas aged contrails (aspherical ice crystals and higher RHI values) with more
pristine ice are categorized in clusters 3 and 4, and finally natural cirrus (low NO concentrations) are
found in cluster 5. A less precise analysis (using onboard camera) reveals that cluster 0 corresponds
essentially to the primary wake created below the secondary wake behind an aircraft. These different
clusters are defined arbitrarily according to ATC information and according to their optical
differences through the three first principal components. In the following, the clusters will be
referenced according to the contrail evolution stage
-  Cluster 0 : Primary Wake (PW)
-  Cluster 1 : Young Contrail 1 (YC1)
-  Cluster 2 : Young Contrail 2 (YC2)
-  Cluster 3 : Aged Contrail Clean (ACC)
-  Cluster 4 : Aged Contrail (AC)
-  Cluster 5 : Cirrus Cloud (CC).
Thus, the next step is to validate these cluster definitions according to the different tracers. One has
to keep in mind that some points are still arbitrarily attributed to a particular cluster without strong
physical justification.
**3.2.2 Application to other CONCERT flights**
In this section we investigate the possibility to complement the previous analysis with additional
cloud optical measurements performed during other CONCERT flights to increase the robustness of
the method.

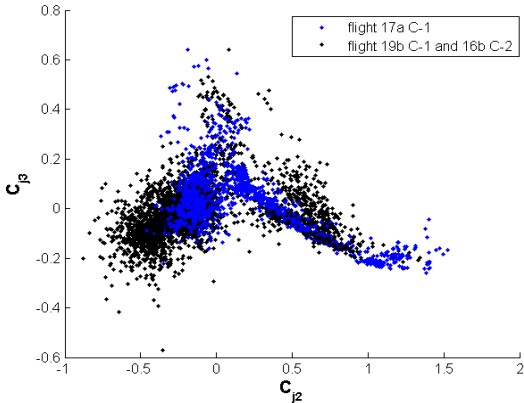

Figure 4: Example of data projection in the $C_{j2}/C_{j3}$ space where data
from flight 17a (blue data points) are superimposed on the data from
the benchmark flights 19b and 16b (black data points).

The principal components obtained on the basis of the measurements performed during flight 16b
and 19b are considered as our reference axes. Now, the ASC measured during other flights can be
projected on this space of principal components. The coordinates of these flights are calculated from



Eq. (4). An example of this data projection is illustrated in Fig. 4 where flight 17a is represented in
the $C_{j2}/C_{j3}$ space. Every PCA data point can be attributed to one cluster previously defined by the
PCA implemented with flights 16b and 19b data (black points). In other words, every ASC measured
during another flight can be merged (projected) into the expansion coefficient diagram relative to the
base measurements performed during flights 16b and 19b. Data points sharing similar optical
properties will still be close to each other. According to the cluster definition, Fig. 4 shows that
different contrail phases have been experienced during flight 17a. Data points are mostly grouped
into cluster ACC, but are also present in clusters AC, YC2, and PW. Finally, cloud data encountered
during this flight are mainly categorized as young and aged contrails.
We follow this methodology to project and classify each additional "contrail" flight performed
during both CONCERT campaigns. In order to attribute each measurement (or point) to one cluster
and to enhance the statistical significance of our clustering analysis we compute the Mahalanobis
distance (De Maesschalck et al., 2000). Clusters are defined by their means (or centers), standard
deviations (or widths), and cross-correlations (or tilts). The Mahalanobis distance is given by the
equation:

$$D_M(x)_i = \sqrt{(x - \mu_i)^T S_i^{-1}(x - \mu_i)} \qquad (5)$$

with $D_M$ the Mahalanobis distance between point $\chi$ and the $i^{th}$ cluster center, $\mu_i$ the N-dimensional
mean of this cluster and $S_i$ its covariance matrix. Each data point can be associated to a specific cluster
corresponding to the shorter Mahalanobis distance, and the ellipsoids' eccentricity and width can be
adjusted.

| | | | Cluster | | | | | | Number of points | Age (s) |
| | | | PW | ACC | AC | YC1 | YC2 | CC | | |
| | | | 1st wake | aged contrail | | young contrail | | Cirrus | | |
| Day / Aitcraft | 17a C-1 | TOTAL | | | | | | | 1435 | |
| | | A340-311 | | | | | | | 359 | 61 - 144 |
| | 17b C-1 | TOTAL | | | | | | | 2715 | |
| | | B737-500 | | | | | | | 310 | 77 - 151 |
| | | A340-642 | | | | | | | 100 | 82 - 139 |
| | | NC | | | | | | | 189 | - |
| | 19a C-1 | TOTAL | | | | | | | 2152 | |
| | | A319-111 | | | | | | | 628 | 94 - 129 |
| | | A340-311 | | | | | | | 175 | 63 - 90 |
| | 19b C-1 | TOTAL | | | | | | | 1647 | |
| | | B767-300 | | | | | | | 319 | 77 - 107 |
| | | CRJ-2 | | | | | | | 151 | 80 - 95 |
| | | A380-841 | | | | | | | 677 | 109 - 240 |
| | 20 C-1 | TOTAL | | | | | | | 1434 | |
| | | B737-300 | | | | | | | 64 | 90 - 290 |
| | 16b C-2 | TOTAL | | | | | | | 1511 | |
| | | A340-600 | | | | | | | 128 | 100 - 132 |
| | | B777 | | | | | | | 378 | 120 - 160 |
| | | A321 | | | | | | | 135 | 70 - 95 |
| | 17 C-2 | TOTAL | | | | | | | 2904 | |
| | | NC1 | | | | | | | 498 | - |
| | | NC2 | | | | | | | 233 | - |
| | 24 C-2 | TOTAL | | | | | | | 1380 | |
| | | B777 | | | | | | | 371 | 112 - 178 |

Table 1: Classification relative to the six clusters on the Cj2/Cj3 representation of the PCA of all data points for each flight of the two CONCERT campaigns (C-1 in November 2008 and C-2 in September 2011). The length of the bars represent the relative contribution of data points of individual contrails (blue bars) and also entire flights (black bars) to the 6 individual clusters.

The classification relative to the six clusters shown on the $C_{j3}$ vs $C_{j2}$ and the $C_{j1}$ and $C_{j2}$ expansion
diagrams is summarized in Table 1. A total of 8 flights (6 additional flights) representing 4426 ASC
measurements was processed. The lengths of the bars in Table 1 represent the relative contributions
of data points to the different clusters: a) black bar merge cloud data points (with extinction coefficient



higher than 0.1 km$^{-1}$) for entire flights and b) blue bars present individual aircraft contrails within specific flights. An important fraction (at least more than 30%) of data points is detected in clusters ACC and CC for each flight during the two campaigns. This indicates that these data points are sampled in aged contrail and sometimes natural cirrus. For flights more clearly performed in well visible contrails outside natural cirrus (earlier development stage and/or intensified persistent elder contrails), significant fractions of data points are associated to clusters PW, YC1, and YC2 (young contrails) for both CONCERT-1 and CONCERT-2 campaigns. However, these flights are also characterized by a significant contribution of data points to cluster ACC (aged contrails clean) and to a minor extent in cluster AC (aged contrails, mostly corresponding to measurements performed during two different B777 contrail chasing events).

These results are in reasonable agreement with previous conclusions (this subsection) about cluster definitions and associated contrail / ice cloud characteristics. Very young contrails have been chased during CONCERT-1 (flights 19a, 19b and 20) and during CONCERT-2 (flights 16b, 17 and 24). Another interesting result is related to flight 17 during CONCERT-2 (flight 17 C-2). No contrail information has been communicated from ATC, however the Falcon has been flying apparently in visible contrails, probably too old for ATC recognition. Data points can mainly be attributed to cluster CC, and to a minor extent to cluster ACC and cluster YC2. This observation suggests that significantly aged contrails have been sampled, resembling strikingly natural cirrus clouds. Indeed, crystal formation and growth processes in contrails and natural cirrus suggest that very old contrails more and more resemble natural cirrus properties. From Table 1 it is obvious that an important amount of data points had been sampled in natural cirrus during this flight. All these natural cirrus data points appear in the black bars but only to a minor extent in the blue bars limited to ATC communicated contrail sequences.

ATC information on contrail ages has been collected during each chasing. Some chasings have been performed less than 100 s after contrail formation. This is the case for the A340 contrail during flight 19a and for the CRJ-2 contrail during flight 19b of CONCERT-1 and for the A321 contrail during flight 16b of CONCERT-2. One can notice that the contrail ages are well correlated to chosen cluster definitions, revealing that contrail data relative to the A340 are included in cluster YC1 and YC2 (young contrails) for more than 53% of the data points, and nearly 65% for the CRJ-2 and 88% for the A321. According to our cluster classification, only 12% of the data points gathered during these three flights correspond to aged contrail (cluster ACC and AC) categories in contrast to other CONCERT-1 and CONCERT-2 flights (with more than 30% of data points associated to ACC and AC). Even though it is still difficult to associate contrail ages to measurement points, the "contrail age" ranges are in agreement with the cluster definitions.

## 4 Evolution of contrail properties

### 4.1 Optical and chemical cluster properties

As demonstrated in the previous part, cloud events can be separated according to their light-scattering properties. Six clusters were defined based on two flights with significant number of data points for each cluster. In this section we present mean optical, chemical, and microphysical properties for each of the six clusters. These mean properties have been calculated over all data points associated to the 6 individual clusters (all flights, both CONCERT campaigns). Figure 5a, 5c, and 5d illustrate the normalized frequency distributions of the asymmetry parameter (gPN), the extinction coefficient (ExtPN), and NO concentrations for the six clusters, respectively. Figure 5b represents mean normalized scattering phase functions, also for the 6 clusters.



The asymmetry parameter gPN statistics for the six clusters shown in Fig. 5a provide the most
relevant information on cloud characteristics and the related context of contrail evolution/age. In
agreement with findings in Gayet et al. (2012), aged contrails (cluster ACC and AC) and natural
cirrus (cluster CC) correspond to gPN values between 0.72 and 0.80, younger contrails (cluster YC1
and YC2) to gPN between 0.80 and 0.86, and primary wake measurements (cluster PW) to gPN above
0.86. This result can be explained with the time evolution of ice crystal shapes after exhaust from
quasi-spherical ice particle to, e.g. column, needle, bullet, and bullet-rosette type crystals. In the
primary wake, the pressure increases associated in the descending vortex. This leads to adiabatic
heating and subsequent sublimation processes of the ice crystals (Lewellen and Lewellen, 2001;
Unterstrasser et al., 2016) and explains a return to spherical shapes and high values of the asymmetry
coefficients.

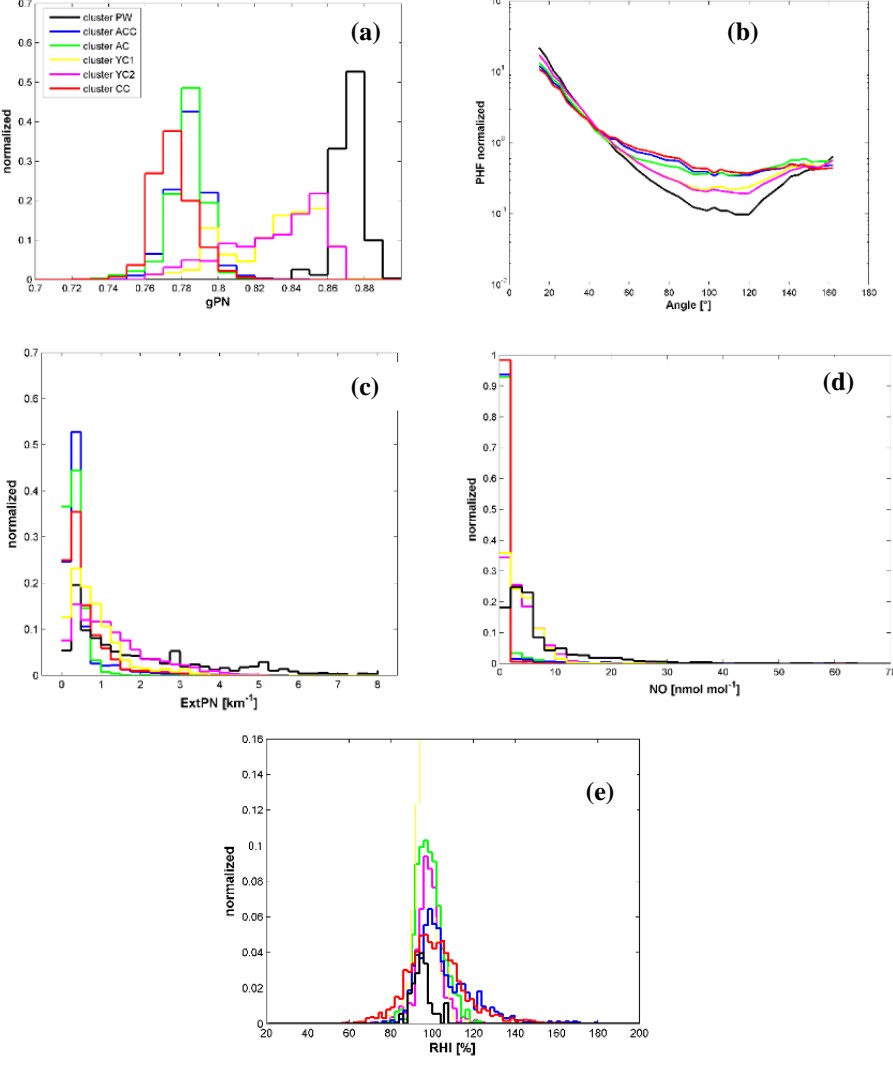

Figure 5: Normalized histograms of a) asymmetry coefficient, b) phase function, c) extinction retrieved by
Polar Nephelomter, d) NO concentration for all flights, and e) RHI conditions for CONCERT-2 flights.





The normalized phase functions are presented in Fig. 5b. Primary wake phase functions (cluster PW) are clearly different from young contrail phase functions (cluster YC1 and YC2), which are themselves different from aged contrails (cluster ACC and AC) and natural cirrus (cluster CC) phase functions. The main difference between the averaged phase functions is observed for the side scattering region (50°-140°) which is related to changes of ice particles shapes and to the proportion of spherical ice crystals within the contrails. This behavior is expected and also in agreement with position of cluster PW, YC2 and YC1 on the expansion coefficient diagram (Fig. 2). The decrease of the $C_{j2}$ coefficient is associated to a side scattering enhancement. Therefore, very young contrails composed of a majority of spherical ice crystals are characterized by phase functions with a substantial scattering at forward angles associated with much lower scattering at sideward angles. As the contrails evolve these features smooth out leading to phase functions with a featureless behavior and a more flat appearance at side scattering angles. Finally, the averaged normalized phase functions of old contrails and natural cirrus are resembling each other. This also explains that they are difficult to discriminate with the PCA.

The extinction coefficient statistics are presented in Fig. 5c. All the aged contrails (cluster ACC and AC) exhibit extinction coefficients lower than 2 $km^{-1}$. The same statement applies for 80% of the sampled natural cirrus (cluster CC). For younger contrails (cluster YC1 and YC2) extinction coefficients can reach 5 $km^{-1}$. Largest extinction coefficients are achieved in primary wake measurements sorted into cluster PW with extinction coefficients reaching values up to 8 $km^{-1}$. Still, the main fraction (more than 50% of data points) of young contrail data yields extinction coefficients between 0 and 1 $km^{-1}$.

Concentrations of chemical species also allow characterizing contrail/cirrus cloud data. The concentration depends strongly on the type of the pursued aircraft. Figure 5d shows mean concentrations of nitrogen oxide NO data points attributed to the six individual clusters. Young contrail NO concentrations (cluster PW, YC1 and YC2) can reach values up to 10 nmol $mol^{-1}$ and up to 60 nmol $mol^{-1}$ in the primary wake. In contrast, in aged contrails and in natural cirrus (cluster ACC, AC and CC) NO concentrations do not exceed 10 nmol $mol^{-1}$ (which is true for more than 97.5%, 99.6%, and 99.7% of data points for clusters ACC, AC, and CC, respectively). Indeed, after exhaust, concentrations of nitrogen oxide NO and sulfur dioxide $SO_2$ created by combustion reactions decrease rapidly due to the dispersion in the upper troposphere and reactions with other molecules.

Finally, saturation conditions with respect to ice are presented in Fig. 5e for all clusters and for CONCERT-2 flights only. The predominant measured ambient relative humidity in all clusters is around 95%. Cluster ACC and CC (blue and red lines) show higher RHI values (more than 120%) than other clusters. Thus, this can explain the formation of natural cirrus and persistent contrails for these ambient conditions. Contrary to all other clusters, no supersaturation is observed for cluster PW (in black), defined as primary wake measurements. This result is in agreement with the definition of the primary wake, which is still in the non-persistent phase of the contrail.

The above results highlight that the principal component analysis, based on the ASC measurements described in Sect. 3, allows to discriminate different types of contrails. Specific optical and chemical properties can thus be characterized for each contrail phase and can be related to their evolution. An interesting aspect is that the PCA analysis facilitates to connect clusters of optical properties to microphysical characteristics of the contrails within specific clusters.

### 4.2 Microphysical cluster properties

Microphysical properties are assessed using the combination of FSSP-300 and 2DC measurements. They have been analyzed for each cluster for hydrometeor diameters between 0.5 µm and 800 µm.





Figure 6 shows mean volume particle size distributions (PSD) for all six clusters measured from the
FSSP-300 and 2DC with respective limited instrumental size ranges. A linear interpolation in
logarithmic space has been applied for each PSD in the size range from 17 µm to 70 µm which is not
accurately documented by the two instruments. These results include all flights of the study (8 flights
from CONCERT-1 and 2). It is important to note that more than 500 data points are included in each
cluster with a maximum of 6300 data points for cluster ACC.

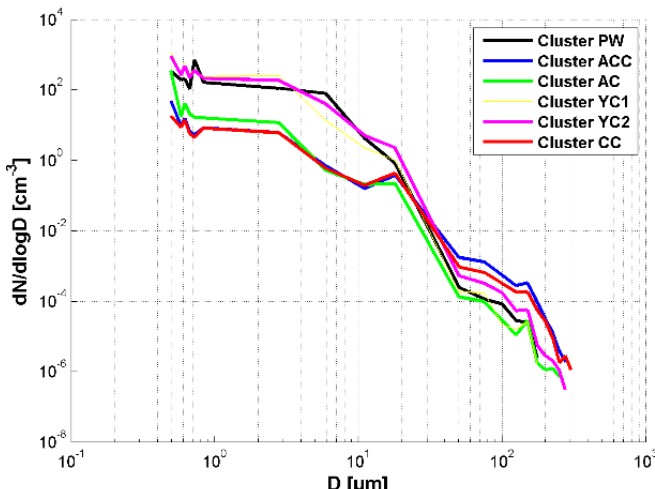

Figure 6: Number particle size distributions for each cluster including all data points of all flights.
FSSP-300 measurements from 0.5 to 17 µm and 2DC measurements from 70 µm to 800 µm. The data
are linearly interpolated in logarithm space between 17 µm and 70µm.

Figure 6 shows that the mean number PSDs for each cluster are consistent with the cluster definition.
Indeed, two categories of PSD can be observed.  Within the FSSP-300 size range, PSD relative to old
contrails (cluster ACC and AC) and cirrus (cluster CC) exhibit more than one order of magnitude
lower number concentrations of small ice crystal compared to young contrails (cluster YC1 and YC2)
and primary wake (cluster PW). Within the two groups, no significant differences can be noticed due
to uncertainties of the FSSP-300 number concentration measurements of 30% for concentrations
around 5cm$^{-3}$ and 75% for concentrations around 0.5 cm$^{-3}$ (Gayet et al., 2002). The differences
between these two groups can been explained by the production of small ice crystals in fresh exhaust
plumes followed by rapid dilution during subsequent minutes after the exhaust. Within the 2DC
range, the PSDs are also in agreement with the cluster definitions. A higher concentration of large ice
crystals with diameters around 100 µm and beyond are expected for natural cirrus and significantly
well-developed contrails. This is particularly well illustrated on the PSD in the 2D-C size range where
a higher concentration of large ice crystal is observed for clusters ACC and CC compared to the
younger contrails. However, these PSD do not allow discriminating young contrails in primary wake
(cluster PW) from contrails in the secondary wake (cluster YC1 and YC2).





| Extinction (km⁻¹) | | Mean | std | Mediane | prctile 25 | prctile 75 |
|---|---|---|---|---|---|---|
| cluster | PW | 3,904 | 4,386 | 1,972 | 0,528 | 6,125 |
| | ACC | 0,088 | 0,282 | 0,028 | 0,009 | 0,057 |
| | AC | 0,079 | 0,186 | 0,024 | 0,001 | 0,071 |
| | YC1 | 2,037 | 2,363 | 1,307 | 0,440 | 2,667 |
| | YC2 | 2,163 | 2,816 | 1,227 | 0,316 | 2,634 |
| | CC | 0,113 | 0,237 | 0,065 | 0,030 | 0,127 |

| IWC (mg m⁻³) | | Mean | std | Mediane | prctile 25 | prctile 75 |
|---|---|---|---|---|---|---|
| cluster | PW | 15,46 | 21,56 | 6,26 | 0,87 | 22,79 |
| | ACC | 8,74 | 37,77 | 0,35 | 0,01 | 3,33 |
| | AC | 1,65 | 19,76 | 0,02 | 0,00 | 0,22 |
| | YC1 | 5,29 | 11,45 | 1,46 | 0,17 | 5,78 |
| | YC2 | 7,85 | 23,40 | 1,36 | 0,14 | 7,77 |
| | CC | 28,69 | 145,73 | 0,96 | 0,05 | 2,87 |

| NTOTAL (cm⁻³) | | Mean | std | Mediane | prctile 25 | prctile 75 |
|---|---|---|---|---|---|---|
| cluster | PW | 125,94 | 98,37 | 109,54 | 54,24 | 166,57 |
| | ACC | 5,57 | 17,86 | 1,29 | 0,76 | 2,41 |
| | AC | 17,89 | 33,48 | 2,92 | 1,21 | 32,90 |
| | YC1 | 155,80 | 159,42 | 106,65 | 38,43 | 207,70 |
| | YC2 | 164,20 | 173,17 | 103,81 | 42,26 | 211,85 |
| | CC | 6,06 | 10,12 | 3,75 | 2,17 | 6,81 |

Table 2: Optical and microphysical properties for each cluster according interpolated particle size distributions from FSSP-300 and 2DC measurements.

Table 2 presents ice water content (IWC, in mg m⁻³) and total number of ice crystals (NTOTAL, in
particles cm⁻³) derived from the measured PSD for each cluster. The extinction coefficient (in km⁻¹)
obtained from the PN measurements is also displayed. Despite the large uncertainties associated to
both instruments and the interpolation method (for ice crystals with diameters ranging from 17 µm
and 70 µm), these results again prove that each cluster can be connected to specific contrail phases.
The microphysical and optical properties of cluster PW are in agreement with the cloud properties
excepted in the primary wakes. The extinction coefficient has a mean value of 3.9 km⁻¹, IWC is close
to 15.5 mg m⁻³, and the number concentration yields a typical value of 125 particles cm⁻³. Young
(clusters YC1 and YC2) and aged contrails (clusters ACC and AC) exhibit distinctive differences in
their optical and microphysical properties. Higher extinction coefficients and ice number
concentration, 2 km⁻¹ and 160 cm⁻³, respectively, characterize young contrails compared to aged
contrails with 0.08 km⁻¹ and 10 particles cm⁻³, respectively. Cluster CC corresponds to natural cirrus
clouds where significant atmospheric spreading and ice growth occurred. Thus, within this cluster the
extinction coefficients (mean values of 0.1 km⁻¹) as well as the number concentration of ice crystals
(around 6 particles cm⁻³) are very low. The IWC is higher with a mean value of 28.7 mg m⁻³ due to
ice crystal growth in supersaturated conditions.
However, it is difficult to discriminate young contrail cases (YC1 and YC2) based on their
microphysical properties. Clusters ACC and AC microphysical properties are also similar but ACC
IWC and number concentrations are closer to the ones of the cirrus case indicating a more evolved
stage of the observed ACC contrail cluster.
**Conclusions**
In this study, a new form of statistical analysis of contrail to cirrus evolution is presented based on
two intensive contrail measurement campaigns, CONCERT-1 and CONCERT-2. The data are used



to study optical and microphysical properties of contrails during their evolution from young contrails to contrail-cirrus clouds, and ambient natural cirrus clouds. The combination of optical, microphysical, and chemical airborne measurement data was used to present an extended view of cloud properties, and to merge those results with ATC flight information about sampled contrails.

A Principal Component Analysis (PCA) methodology has been applied to the measured Polar Nephelometer scattering phase function data in order to facilitate a distinction between cloud properties in different contrail phases. The PCA results were derived first for two reference flights that sampled contrails and cirrus in various development stages, including the primary wake, the young secondary wake, old contrails (few minutes after formation) and natural cirrus. For these flights, the PCA clearly demonstrates the potential to separate different groups of clouds, justifying the use of these two flights as a benchmark. The scattering phase functions measured during other CONCERT flights were then projected into the space of principal components obtained from the two reference flights. Mahalanobis distances were used to measure the separation between the additional data points and the data in the predefined clusters. From the entire data set, the cloud properties in the various contrail development stages can be separated and analyzed separately. The analysis demonstrates that the clearest separation between clusters is related to particle shape, which is significantly controlling the scattering phase function and the derived asymmetry parameter gPN. The asymmetry parameter clearly separates young contrails (gPN of 0.72 to 0.80) from contrail/cirrus with gPN ranging from 0.80 to 0.88. Since it is still difficult to evaluate the exact age of each measurement, young and aged contrails are classified also by their optical and chemical properties. The measured NO concentrations are useful to distinguish natural cirrus from old contrails. Despite the important gap between the two instruments used to measure particle size distributions, particle size spectra and related mean values of the ice particle number concentration, extinction and ice water content have been determined for each cluster. The various clusters clearly show different size distributions. In good agreement with previous findings on optical and chemical properties, we find that young contrails have more than a factor of ten higher number concentrations of small ice crystals (with diameters lower than 20 µm) than aged contrails and natural cirrus. On the other hand, aged contrails and natural cirrus contain larger ice crystals, with diameters larger than 75 µm. The optical and microphysical properties of the aged contrail cirrus are often similar to those found in the ambient "natural" cirrus clouds. The results show that the PCA method allows to identify and discriminate different contrail growth stages and to provide an independent method for the characterization of the evolution of contrail properties.

Accurate modeling of cirrus or contrails' single scattering properties is a primary condition for the interpretation of remote sensing measurements. Therefore, measurements of the optical characteristics of ice crystals in natural conditions are still needed for validation of numerical techniques and for the determination of free parameters in light scattering models. In this context, the results from the PCA could be used to develop representative parameterizations of the scattering and geometrical properties of the ice crystals' shapes and sizes observed in the visible wavelength range that then have to be extrapolated into the near infrared.

**Acknowledgments**

We thank financing by the Helmholtz Association under contract VH-NG-309 and W2/W3-60. Part of this work was funded by DFG SPP HALO 1294 contract VO1504/4-1, and by the DLR project Eco2Fly in ML-CIRRUS-cirrus special issue. We thank Lufthansa, the DLR flight department and DFS for excellent support during the campaign. The in-situ data can be found in the HALO-database (https://halo-db.pa.op.dlr.de/).

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
