# Peer review of "Statistical Analysis of Contrail to Cirrus Evolution during the Contrail and Cirrus Experiments (CONCERT)"

_Atmospheric Chemistry and Physics, 2017_

## Referee Comment (RC1) · Anonymous Referee #1 · 20 Nov 2017

Review of

**Statistical Analysis of Contrail to Cirrus Evolution during
the Contrail and Cirrus Experiments (CONCERT)**

by  Chauvigné et al.

The manuscript presents a new form of statistical analysis, the Principal Component Analysis (PCA), to investigate contrail to cirrus evolution based on two field campaigns.  The observed ice clouds are divided in six clusters representative of different development stages of the contrails (primary wake, young contrail, contrail-cirrus and natural cirrus). Optical, chemical and microphysical properties of the clusters are then characterized to describe the ice-cloud properties during  contrail to cirrus evolution .

Overall,  the paper is very interesting and the topic is timely and suitable for ACP.  Especially, the new approach to distiguish contrail and cirrus clouds seems to be promising.   The manuscript is well structured and fluently to read.   I found a number of minor points that I think needs claryfication before publishing the manuscript, they are listed below. Nevertheles, my overall rating is major revisions because of two points emphasized here:

– I strongly recommend to include the RHi measurements of flight 19b during CONERT 1.
  They are published in Gayet et al. (2012), so the data are available – see comment 8).

– Some numbers in Table 2 needs to be checked,
     the mean and median  IWC and, especially,
     the mean and median of Ntotal for CC are too high for natural cirrus – see comment 20).

**Comments:**

1) Introduction: I like the very detailed  introduction, but recommend to introduce more subsections (maybe even with titles), since now there are quite long paragraphs and the structure is not clearly visible.

2) Page 5, Line 194-195:  *‚Particle size distributions and corresponding microphysical and optical integrated properties (IWC, Deff, N, and extinction) were derived from FSSP-300 measurements (Baumgardner et al., 1992).‘*

FSSP measurements does not include the larger ice particle and is thus know to be not suitable for calculations of at least IWC and Deff.  The 2DC was also flown during the field campaign, so why not combining the two probes for the calculations ?  The missing size range between the two probes could be interpolated.

3)  Shattering of large ice crystals is negligible in contrails since the maximum size of the crystals is not large enough to cause this effect. I would mention that somewhere in the manuscript.

4)  Page 5, lines 220-221: *‚The bulk parameters were calculated assuming the surface-equivalent diameter relationships of Heymsfield (1972) and Locatelli and Hobbs, (1974).‘*

Which  bulk parameters do you mean ?

5) Page 6, line 234 - 238 :  Calculation of  $IWC_{non\text{-}spherical}$:

Is the validity of  Equ. (6) ever  checked by comparison to bulk IWC measurements?

6)  Page 6, lines 254-256: ‚*In addition, hygrometers using the Lyman-alpha technique*
     *(FISH, Zöger et al., 1999; Meyer et al., 2015), and frost point hygrometers*
     *(CR-2, Heller et al., 2017) were implemented on the Falcon during CONCERT-1 and 2.‘*
    Please add the names of the hygrometers as indicated in blue.

7) Figure 1:

a) Caption: ‚*Time series at 1 s resolution for flights a) 19b (CONCERT 1) and*
           *b) 16b (CONCERT 2).‘*
   Please add the names of the campaigns as indicated in blue.

b) Plot of gPN: it would be helpful if a line at 0.85 woul be drawn in the figure to better see if the particles are spherical or aspherical.

c) Plot of NO: a log scale might be better here, especially for Flight 19b from CONCERT 1.

8) Flight 19b from  CONCERT 1:   Why are the RHi measurements of that flight not included here? They are published in Gayet et al. (2012), so the data are available.

I strongly recommend to include this data. It can be  seen later in the paper that the number of RHi data from only flight 16 b fom CONCERT 2 is too low to apply the PCA analysis, see Figure 3, bottom right.  Further, on page 8, lines 304-305 you write for flight 16 b: ‘*However, no accurate ambient RHI data can retrieved for measurements in natural cirrus due to instrumental calibration problems*.‘ but there are  natural cirrus data available for   19b , CONCERT 1, yes ?

In addition, on page 6, line 252 you state the importance of RHI to characterize contrail ice crystals and on page 8, 3[rd] paragraph, you describe how RHI influences the sphericity of ice crystals. So  I  think it is of  importantance not to leave out  available RHI measurements!

9) Page 8,  lines 294-297: ‚*In a supersaturated environment of contrails, crystals grow by water vapor deposition and become increasingly aspherical with time. This is why spherical ice crystals prevail in very young contrails with an asymmetry coefficient around 0.85 with RHI above 100%.‘*

These sentences are a bit confusing. The reason that the ice crystals in young contrails are spherical under supersaturated conditions is that the time was too short to become aspherical, yes ?  Maybe better:  ‚*In very young contrails, not enough time has passed so that despite RHI is above 100% spherical ice crystals with an asymmetry coefficient around 0.85 prevail.‘*

10) Page 8, lines 305-309: ‚*A good example of the evolution of gPN is the CRJ-2 contrail observed between 11:40 and 11:45 during flight 19b. The sequence illustrates the potential of the gPN measurement to characterize the evolution of contrail properties, with decreasing crystal sphericity documented by the decreasing asymmetry parameter from 0.88 to 0.79 (uncertainties around 0.04) after only 5 min and down to 0.77 after 20 min.‘*

Again, it would be very good to see the corresponding RHI measurmenst here.

11) Page 8/9, last/first paragraph:   Correlations between parameters are hard to recognize from  Figure 1.  Scatterplots for the main correlating parameters (gPN, RHI - from both flights, NO, extPN) would greatly improve the visualization of the discussed relations.

12) Page 12, last paragraph:   For a  better understanding of this paragraph, I recommend
    → to make a table of the of the cluster numbers and the corresponding definitions (now listed at page 13, lines 453 – 458) and refer to the table at the beginning of the paragraph. In the present form, it is hard to follow the text without knowing the meaning of the numbers.
    → Further, it would be good to note the abbrevations of the numbers (0: PW, 1: YC1, 2: ...) in one panel of Figure 3, e.g. in 3a.

13) Page 12, line 420-421:
    ‚Figure 3a suggests an increase of Cj2 and a decrease of Cj1 with increasing aircraft size.‘

    In Fig. 3a  Cj2 vs. Cj3 is plotted, in the text you refer Cj2 vs. Cj1 – please correct.

14)  Definition of Clusters 3 (AC: Aged contrail) and 4 (ACC: aged contrail clean):
    What is the difference between the two clusters ? Does ‚clean‘ means low NO ?
    Please explain.

15) Figure 4:  It would be helpful if you would include the circles from Figure 3 (a)
             in this plot.

16)  Page 14, line 479:   attribuate → attribute

17) Figure 5:  I found it difficult to recognize the message of the panels of Figure 5. Here are
    some recommendation how this important figure can be improved:
    a)  in panel (a),  a vertical line at gpN=0.85 would be helpful to distinguish between
              spherical and aspherical.
    b)  in panel (c),  when using a logarithmic scale for the frequency the effects you
              discuss in the text will become better visible.
    c)  in panel (d), a logarithmic scale for NO and also the frequency would help to
              better see the the differences between the clusters.
    d)  in panel (e), a vertical line at RHI=100% would be good.
       Further, in the text it is mentioned that the most frequent value of RHI is 95%.
       Shouldn‘t that be 100% ?  And, the histogram is divided into small RHI intervals
       (2% ?), but the accuracy of the measurements is not better than 10-20%, I guess.
       I recommend to divide RHI in  intervals corresponding to the accuracy and center
       them around 100%.
    e) How many data points does each cluster contain ? This can be indicated in the legend.
    f) The legend could be  included in each panel – this would make it easier for the reader
       to assign the colors to the clusters when zooming the Figure on the screen.
       Another idea could be to use more intuitive colors and sort the legend somehow into
       the stages of development, here a suggestion:
       **PW  YC1  YC2  AC  ACC  CC**

18)  Page 18, line 592:
       ‚Figure 6 shows mean *volume* particle size distributions (PSD) for all six clusters.‘

       I see mean number PSDs – dN/dlogD

19)   Figure 6:   The maximum sizes of PW and YC1 are already close to 200 μm, the maximum size of YC2 is close to that of ACC and CC. I would have expected smaller maximum sizes in the PW and YC categories, because ice crystals needs time to grow to larger sizes?
Further, the maximum size of CC is quite small –  Voigt et al. (2017) show maximum sizes of natural cirrus PSDs up to 1000 μm or more ?  See also comment 19 (b).

20)  Table 2:  (a) I suggest to sort the clusters like recommended under Point 16 f).
                        A further suggestion is to sort Ntotal in two size intervals, namely
                        <~30um and >~30um, since the grouping of the clusters cange with size.

          (b) The mean and median values of IWC does not fit to each other.
              For example,  for PW / CC the means are 15.46/28.69 mg/m3, but the
              medians are 6.26/0.96 mg/m3, i.e. the mean of CC is almost twice the mean
              of PW, but the median of CC is much lower than that of PW ?
              Please check all numbers.

          (c) Mean/median of Ntotal for CC are 6.06/3.75 cm$^{-3}$ .
              This is too high for natural cirrus – from Voigt et al. (2017),  I would
              expect something around 0.1 cm$^{-3}$  or even lower.
              Is that an arithmetical error , shattering or could it be that  contrails are
              accidentally attributed to  CC ?  Please clarify!

21)  One last comment:  could you discuss the possibility to use other/more parameters for the PCA? For example, could  Ntotal be included in the PCA ?  Or in case no Polar Nephelometer is on board, but PSD, IWC and NO is available, do you think the analysis would be possible ?

---

## Referee Comment (RC2) · Anonymous Referee #2 · 28 Nov 2017

The authors present aircraft observations of the scattering properties of ice crystals and the trace gas properties sampled inside 17 contrails during two phases of the CONCERT field experiment. While the results presented here are relevant and interesting, the paper has several areas where more explanation is warranted before I can recommend it for publication. For example, some parts of the introduction need to be reorganized.

The most major flaw of the paper which needs to be address is the selection of the clusters. The authors base their cluster classification on a rough examination of the first three principal components in the x-y plane and seem to draw ellipses around

where they "roughly identify" where the clusters are. However, with recent advances in machine learning, there are more objective methodologies for classifying data into clusters, with the most applicable methodology for a feature space of three variables to be k-means clustering. The authors should either better justify why their current ellipses were chosen and why the feature space was used for the PCA, or use automated clustering techniques. Finally, I think a section on how their contrail cirrus observations fit in with past studies is warranted, since the paper lacks much discussion on how their observations fit in with what is already in the literature. I list some other comments below.

Major comments:

Lines 51-91. This paragraph is too long and needs to be reorganized. For example, there is too much detail on how NO from aircraft exhaust is converted into acids that does not really add to the major point that NO interacts with OH to make nitr(ic)ous + sulfuric acid. I also feel that this can really be 3 paragraphs: one about NO interacting with OH to produce acids, one about the contrail production process and one about the contrail aging process.

Line 109-146: I feel that a lot of the individual data points cited here are better suited for an extra section in the paper comparing your contrail observations against past studies. Right now, no link is made to how your categories compare against these past observations and I think such a comparison is needed in order to justify that the range of values that you observe in your clusters correspond to contrails properties that are observed in nature. Therefore, I recommend shortening this paragraph to just briefly explain how the microphysical properties of contrails evolve with time with leaving specific numbers to a later comparison.

Line 223-225: You aren't using the 2DC for calculating IWC though! I don't see why this sentence is needed. However, I think text here justifying why you are not using observations below 70 microns due to the 2DC's limited response time and depth of

field need to be here.

Section 3.2: I think more justification needs to be given for the choice of your feature space for the PCA, since right now it is presented without really linking the feature space to looking for quantities that we expect to vary in differing stages of contrail cirrus. For example, why did you conduct a PCA on the entire scattering phase function instead of just apply clustering to the asymmetry parameter?

Also, why were the clusters manually chosen instead of using automated techniques like k-means clustering?

Lines 505-512: How do you know that you flew in an aged contrail with no verification from ATC? I think the important conclusion here is more that, microphysically, aged contrails and cirrus are very similar and are difficult to distinguish with this data alone.

Line 518-522: I think this analysis can be better supported by showing the distributions of contrail ages from ATC.

Line 593-595: I would not interpolate data in this range since the interpretation of extrapolated data could be quite dangerous. I would simply state that concentrations in this size range are too uncertain to report due to the 2DC's poorly characterized depth of field and response time.

Lines 607-610: Your YC1 contrails seem to have roughly similar 2DC number concentrations to the aged contrails. Why is that?

Lines 640-667: I would convert this into a bulleted list of conclusions to make this paragraph easier to read.

Figures/Tables:

Figures 5c,d: A logarithmic x-axis would make the lines easier to distinguish.

Figure 6: I would advise removing the lines where you don't have the PSD from the two probes in the ~20 to 70 micron range. Can you also add size distributions from past

studies and include them in the comparison?

Table 2: I think some data from contrails sampled in past studies should be shown and comapred against here and in the paragraph discussing Table 2.

[Figure]

---

## Author Comment (AC1) · 15 Mar 2018

We would like to thank the reviewer for his interesting and constructive suggestions. We have tried to follow every suggestion in order to improve the manuscript. Each reviewer's comments are addressed and the manuscript has been modified accordingly.

Please also note the supplement to this comment: https://www.atmos-chem-phys-discuss.net/acp-2017-946/acp-2017-946-AC1-supplement.zip
* * *
[Figure]

2017.

---

## Author Response (AR1)

RC1

Review of
Statistical Analysis of Contrail to Cirrus Evolution during
the Contrail and Cirrus Experiments (CONCERT)

The manuscript presents a new form of statistical analysis, the Principal Component Analysis (PCA), to investigate contrail to cirrus evolution based on two field campaigns. The observed ice clouds are divided in six clusters representative of different development stages of the contrails (primary wake, young contrail, contrail-cirrus and natural cirrus). Optical, chemical and microphysical properties of the clusters are then characterized to describe the ice-cloud properties during contrail to cirrus evolution.
Overall, the paper is very interesting and the topic is timely and suitable for ACP. Especially, the new approach to distiguish contrail and cirrus clouds seems to be promising. The manuscript is well structured and fluently to read. I found a number of minor points that I think needs claryfication before publishing the manuscript, they are listed below.

Nevertheles, my overall rating is major revisions because of two points emphasized here:
– I strongly recommend to include the RHI measurements of flight 19b during CONERT 1. They are published in Gayet et al. (2012), so the data are available – see comment 8).

– Some numbers in Table 2 needs to be checked, the mean and median IWC and, especially, the mean and median of Ntotal for CC are too high for natural cirrus – see comment 20).

We would like to thank the reviewer for his interesting and constructive suggestions. We have tried to follow every suggestion in order to improve the manuscript. Each reviewer's comments are addressed and the manuscript has been modified accordingly.

A significant change in the paper concerns the implementation of an automatic clustering method (k-mean method) to enhance the statistical significance of the contrail phase discrimination. Subsection "*3,2,2 Clustering analyses*" has been added and gives details of the method. After some tests we found that 16 clusters are necessary to classify the patterns revealed by the PCA analysis. To be in accordance with ATC observations, clusters 8 to 16 deduced from the k-mean method (Figure RC2.1.a.) were gathered into two clusters (A and B, Figure RC2.1.b.). Clusters 2 to 5 have also been merged to one single cluster.

[Figure]

Figure RC1.1.: Clustering analyses according k-mean method. a) First step using 16 clusters and b) second step grouping clusters 8 to 16 to two clusters (A and B).

The following table summarizes the correspondence of the clusters defined by the k-mean methods and the cluster's definitions according to ATC information:

| k-mean clusters | Cluster number | definition | name |
|---|---|---|---|
| 1 | 0 | Primary Wake | **PW** |
| 6 | 1 | Young Contrail 1 | **YC1** |
| 2, 3, 4 and 5 | 2 | Young Contrail 2 | **YC2** |
| A | 3 | Aged Contrail 1 | **AC1** |
| 7 | 4 | Aged Contrail 2 | **AC2** |
| B | 5 | Cirrus Cloud | **CC** |

Table RC1.1: Cluster definitions according the k-mean method.

Comments:

1) Introduction: I like the very detailed introduction, but recommend to introduce more subsections (maybe even with titles), since now there are quite long paragraphs and the structure is not clearly visible.

The structure of the introduction has been changed and two subsections were added, namely:
       1.1. Contrail formation and evolution
       1.2. Optical and microphysical properties of contrail phases

Moreover, the description of contrail chemical properties has been shortened.
Modifications:
l.55:" *Several studies in the past have been dedicated to the evolution of concentrations of nitrogen oxide (NO) and sulphur dioxide (SO$_2$) and their oxidized forms (Kärcher and Voigt, 2006 ; Voigt et al., 2006 ; Schäuble et al., 2009 ; Jurkat et al., 2011).*"

2) Page 5, Line 194-195: *Particle size distributions and corresponding microphysical and optical integrated properties (IWC, Deff, N, and extinction) were derived from FSSP-300 measurements (Baumgardner et al., 1992).*
FSSP measurements does not include the larger ice particle and is thus know to be not suitable for calculations of at least IWC and Deff. The 2DC was also flown during the field campaign, so why not combining the two probes for the calculations? The missing size range between the two probes could be interpolated.

Optical and microphysical properties of contrail particles cannot be fully retrieved without the 2DC measurements. In the present study, these properties are retrieved with both instruments, FSSP and 2DC, to consider as much as possible the full-size range of the particle size distribution. The data are linearly interpolated in logarithm space between the two instrument ranges. The text has been clarified according to this remark.

Modifications:
l.191: "*Particle size distributions and corresponding microphysical and optical integrated properties (IWC, Deff, N, and extinction) were derived from both FSSP-300 and 2DC measurements.*"

3) Shattering of large ice crystals is negligible in contrails since the maximum size of the crystals is not large enough to cause this effect. I would mention that somewhere in the manuscript.

Indeed, the shattering of particles with size larger than typically 100µm on the probe inlet can affect the measurements. In addition, this effect can also explain the high number concentrations observed in natural cirrus (Table 3). It is now mentioned in the Manuscript.

Modifications:
l.235: "*These equations do not account for possible shattering of large ice crystals on the probe inlets. This effect is minimized in young contrails but can lead to an overestimation of small ice crystal concentration in contrail cirrus clouds.*"

l.647: "*Besides to interpolation between the FSSP-300 and the 2DC measurements, the assumed shape (spherical or aspherical), and shattering of large ice particles in cirrus and aged contrails can also have a significant effect on the measurement of optical and microphysical properties (Gayet et al., 2012).*"

4) Page 5, lines 220-221: *The bulk parameters were calculated assuming the surface-equivalent diameter relationships of Heymsfield (1972) and Locatelli and Hobbs, (1974).* Which bulk parameters do you mean?

The bulk parameters referred to the IWC, Deff, and extinction. It should be integrated or derived microphysical parameters. It was clarified in the text and, this sentence has been removed and the paragraph modified.

Modifications:
l.221: "*For spherical and non-spherical particles, the extinction coefficients are calculated from the following equation:*

$$Ext = \frac{\pi}{4} \sum_i \beta_{ext}^i N_i D_i^2 \qquad (1)$$

*where $\beta_{ext}^i$ is the extinction efficiency (values depend on spherical or aspherical particle characterization), $D_i$ the mean diameter in channel i, and $N_i$ the number concentration.*

*Different approaches are used to retrieve ice water content from spherical and non-spherical particles (Garret et al., 2003 ; Gayet et al., 2004 ; Gayet et al., 2012). For spherical particles (gPN > 0.85), IWC is computed from the following equation:*

$$IWC_{spherical} = \frac{\pi}{6} \rho_{ice} \sum_i N_i D_i^3 \qquad (2)$$

*with $\rho_{ice}$ the bulk ice density (0.917 g cm$^{-3}$).*"

5) Page 6, line 234 - 238 : Calculation of IWC non-spherical : Is the validity of Equ. (6) ever checked by comparison to bulk IWC measurements?

The method used to derive IWC from spherical and non-spherical particles with equivalent diameter calculation, has been validated with different measurement techniques (FSSP-300 and polar Nephelometer) in the work of Gayet et al. (2012). The IWC derived from the particle size distribution is hindered by the uncertainties related to the size dependent enhancement factor for ice crystals in the inlet of total water instruments. There is a strong particle size dependence of the enhancement factor for bulk water instrument inlets in the small particle size range lower than 20 µm representative for young contrails. Further, there is little information on the particle shape effect on the enhancement factor. Finally, the inlet position of bulk phase instruments near the fuselage of the aircraft may introduce additional ambiguities in the IWC measured near the aircraft fuselage. These uncertainties limit the quality of an assessment of the shape effect on the used mass dimension relationship using a comparison to the IWC measurements derived from bulk phase instruments.

6) Page 6, lines 254-256: *In addition, hygrometers using the Lyman-alpha technique (FISH, Zöger et al., 1999; Meyer et al., 2015), and frost point hygrometers (CR-2, Heller et al., 2017) were implemented on the Falcon during CONCERT-1 and 2.*
Please add the names of the hygrometers as indicated in blue.

The two hygrometers names have been added in the text.

Modifications:
l.248: "*(FISH, Zöger et al., 1999; Meyer et al., 2015), and frost point hygrometers (CR-2, Heller et al., 2017) were deployed on the Falcon during CONCERT-1 and 2.*"

7) Figure 1:
    *a) Caption: Time series at 1 s resolution for flights a) 19b (CONCERT 1) and*
       *b) 16b (CONCERT 2).*
    Please add the names of the campaigns as indicated in blue.

    b) Plot of gPN: it would be helpful if a line at 0.85 would be drawn in the figure to better see if the particles are spherical or aspherical.

    c) Plot of NO: a log scale might be better here, especially for Flight 19b from CONCERT 1.

All these suggestions have been considered. Figure 1 and its legend have been modified to improve the reading of the figure.

8) Flight 19b from CONCERT 1: Why are the RHI measurements of that flight not included here? They are published in Gayet et al. (2012), so the data are available.

RHI measurements performed during Flight 16b show typically values higher than (or close to) 100% when contrails are detected by the PN (extinction >0.1 km$^{-1}$). For Flight 19b, RHI values are always higher than 75% but values higher or equal to 100% are scarce. Moreover, contrail/cirrus events identified by the PN or chemical measurements do not seem to be correlated with RHI values. RHI measurements during Flight 19b and CONCERT 1 in general should be taken with caution. An additional bias in the temperature measurements of the Falcon may be responsible for an offset in the RHI measurements inside and outside of contrails during the CONCERT 1. This was not observed during CONCERT 2 where the temperature sensors have been extensively calibrated and RHI peaks near 100% are found in contrails and natural cirrus (Kaufmann et al., 2014). In addition, the descent of the primary wake within the wake vortices leads to an increase in temperature and thus an altitude dependent RHI profile within the contrails as observed by Gayet et al. (2012), Jeßberger et al. (2013) and Kaufmann et al. (2014). Thus, RHI measurements are shown for both campaigns but had to be analysed carefully taking into account the altitude of measurements and calibrations before flights.

Modifications:
RHI measurements for flight 19b (CONCERT 1) are added Figure 1 and Figure 3d.

I strongly recommend to include this data. It can be seen later in the paper that the number of RHI data from only flight 16b from CONCERT 2 is too low to apply the PCA analysis, see Figure 3, bottom right. Further, on page 8, lines 304-305 you write for flight 16 b: *However, no accurate ambient RHI data can retrieved for measurements in natural cirrus due to instrumental calibration problems.* but there are natural cirrus data available for 19b, CONCERT 1, yes?

We apologize for this possible misunderstanding. The PCA is solely based on light scattering measurements performed by the Polar Nephelometer. Other parameters such as RHI or NO concentration are used to validate or evaluate clusters/patterns revealed by the statistical analysis. On the principal component plots (Figure 3), "Natural cirrus" measured during CONCERT 1 are identified based on their scattering properties. Moreover, only one natural cirrus event was observed during flight 16b (none during flight 17) between 17:00 and 17:30. Unfortunately, no RHI measurements were performed during that time.

In addition, on page 6, line 252 you state the importance of RHI to characterize contrail ice crystals and on page 8, 3 rd paragraph, you describe how RHI influences the sphericity of ice crystals. So I think it is of importance not to leave out available RHI measurements!

RHI measurements during CONCERT 1 have been added to the manuscript. Instrumental issues reported in previous works (Kübbeler et al. (2011); Gayet et al. (2012); Jessberger et al. (2013) ; Schumann et al. (2013)) are also mentioned.

Modifications:
l.255: "*RHI measurements during flight 19b as well as instrument shortcomings are discussed in details in Kübbeler et al. (2011), Gayet et al. (2012), Jessberger et al. (2013) and Schumann et al. (2013).*"

9) Page 8, lines 294-297: *In a supersaturated environment of contrails, crystals grow by water vapor deposition and become increasingly aspherical with time. This is why spherical ice crystals prevail in very young contrails with an asymmetry coefficient around 0.85 with RHI above 100%.*

These sentences are a bit confusing. The reason that the ice crystals in young contrails are spherical under supersaturated conditions is that the time was too short to become aspherical, yes ? Maybe better: *In very young contrails, not enough time has passed so that despite RHI is above 100% spherical ice crystals with an asymmetry coefficient around 0.85 prevail.*

Indeed, the previous sentence was not that clear. We have modified the sentence following the reviewer suggestion.

Modifications:
l. 287: "*In a supersaturated environment, crystals grow by water vapour deposition and become increasingly aspherical with time. However, in very young contrails, spherical ice crystals with an asymmetry coefficient around 0.85 prevail.*"

10) Page 8, lines 305-309: *A good example of the evolution of gPN is the CRJ-2 contrail observed between 11:40 and 11:45 during flight 19b. The sequence illustrates the potential of the gPN measurement to characterize the evolution of contrail properties, with decreasing crystal sphericity documented by the decreasing asymmetry parameter from 0.88 to 0.79 (uncertainties around 0.04) after only 5 min and down to 0.77 after 20 min.*
Again, it would be very good to see the corresponding RHI measurements here.

RHI measurements for flight 19b are now added to Figure 1. We can see that despite the presence of a persistent contrail probably due to a supersaturated ambient air, RHI values during this period are too low. Thus, this example shows how the RHI measurements during CONCERT 1 cannot be considered to understand or detect contrail formation.

Modifications:
l. 294: "*However, it is important to note that the RHI measurements during the CRJ-2 chasing events do not show supersaturated conditions, whereas contrail seems persistent. Indeed, RHI measurements should be discussed carefully for this campaign due to calibration issues.*"

11) Page 8/9, last/first paragraph: Correlations between parameters are hard to recognize from Figure 1. Scatterplots for the main correlating parameters (gPN, RHI - from both flights, NO, extPN) would greatly improve the visualization of the discussed relations.

[Figure]

Figure RC1.2.: Correlation between optical and chemical properties during CONCERT 1 and 2 measurement campaign. Colours refers to ATC information for the different chasing and natural cirrus measurements.

The purpose of figure 1 is to illustrate the correlation/correspondence between the main contrail optical and chemical composition trends with ATC information. The only objective of this figure is to help the reader to understand how cloudy and clear sky conditions were determined and to show that parameters like the extinction, the asymmetry parameter and NO concentration can used to identify contrails and cirrus events.

To respond to the reviewer's comment, correlations between extinction, RHI, NO concentration and asymmetry parameter are shown on Figure RC1.2. for 19b and 16b flights. No clear correlations can be seen from these representations since each contrail stages are mixed together. However, NO concentrations and extinction coefficients seem to be linked for particular cases. For clarity, we decided not to show these correlations as it might misguide the reader.

12) Page 12, last paragraph: For a better understanding of this paragraph, I recommend
→ to make a table of the of the cluster numbers and the corresponding definitions (now listed at page 13, lines 453 – 458) and refer to the table at the beginning of the paragraph. In the present form, it is hard to follow the text without knowing the meaning of the numbers.
→ Further, it would be good to note the abbreviations of the numbers (0: PW, 1: YC1, 2: ...) in one panel of Figure 3, e.g. in 3a.

According to reviewer 2 suggestions the clusters have been redefined according Table RC1.1 and abbreviations of the numbers were also added.

Modifications:
The table of cluster definition is added (Table 1). Legends for all subplots has been added in Figure 6 in order to recall each cluster definition.

13) Page 12, line 420-421:
*Figure 3a suggests an increase of Cj2 and a decrease of Cj1 with increasing aircraft size.*
In Fig. 3a Cj2 vs. Cj3 is plotted, in the text you refer Cj2 vs. Cj1 – please correct.

Because the correlation is not as clear as mentioned in the text, this discussion has been removed from the text.

14) Definition of Clusters 3 (AC: Aged contrail) and 4 (ACC: aged contrail clean):
What is the difference between the two clusters? Does ‚clean means low NO?
Please explain.

"Aged contrail" and "aged contrail clean" can be discriminated based on NO concentrations. "Clean" refers to low NO concentrations compared to measurements corresponding to AC conditions. AC and ACC have been replaced by AC1 and AC2 respectively without taking into account NO concentrations.

15) Figure 4: It would be helpful if you would include the circles from Figure 3 (a) in this plot.

Due to the modification of the clustering method, circles in Figure 3 have been removed. Every cluster's centre is now added on the new Figure 4.

16) Page 14, line 479: attribuate → attribute

The recommendation has been added into the all text.

17) Figure 5: I found it difficult to recognize the message of the panels of Figure 5. Here are some recommendation how this important figure can be improved:
a) in panel (a), a vertical line at gpN=0.85 would be helpful to distinguish between spherical and aspherical.
b) in panel (c), when using a logarithmic scale for the frequency the effects you discuss in the text will become better visible.
c) in panel (d), a logarithmic scale for NO and also the frequency would help to better see the the differences between the clusters.
d) in panel (e), a vertical line at RHI=100% would be good.
Further, in the text it is mentioned that the most frequent value of RHI is 95%. Shouldn't that be 100% ? And, the histogram is divided into small RHI intervals (2% ?), but the accuracy of the measurements is not better than 10-20%, I guess. I recommend to divide RHI in intervals corresponding to the accuracy and center them around 100%.
e) How many data points does each cluster contain ? This can be indicated in the legend.
f) The legend could be included in each panel – this would make it easier for the reader to assign the colors to the clusters when zooming the Figure on the screen.
Another idea could be to use more intuitive colors and sort the legend somehow into the stages of development, here a suggestion:
**PW YC1 YC2 AC ACC CC**

All these recommendations have been considered and included in Figure 6. RHI histogram has been modified to include RHI measurements of CONCERT 1.

18) Page 18, line 592:
*Figure 6 shows mean volume particle size distributions (PSD) for all six clusters.*
I see mean number PSDs – dN/dlogD19)

We apologize for this mistake, "volume" has been replaced by "number" into the text l.602.

19) Figure 6: The maximum sizes of PW and YC1 are already close to 200 μm, the maximum size of YC2 is close to that of ACC and CC. I would have expected smaller maximum sizes in the PW and YC categories, because ice crystals needs time to grow to larger sizes?
Further, the maximum size of CC is quite small – Voigt et al. (2017) show maximum sizes of natural cirrus PSDs up to 1000 μm or more ? See also comment 19 (b).

Indeed, no particle measurements have recorded up to 1000 µm during the two campaigns. Voigt et al. (2017) show "natural cirrus" and "contrail cirrus" with mean concentrations higher than 0.01 cm$^{-3}$ for particles diameters around 100µm whereas concentrations do not exceed 0.01 cm$^{-3}$ during CONCERT flights at this particle size range. Because instruments are different between the two campaigns, it can be explained by different shattering effects of largest particles, but also by air speed issues as explained in Febvre et al. (2009). In addition, young contrail particles exhibit diameters up to 200 µm. This effect can also be explained by the detection limit of the 2DC instruments which impacts the concentrations for very low signals (50% and 75% for concentrations of 5 and 0.5 cm-3 respectively, Gayet et al., 2002).
This discussion has been added to the text.

Modifications:
PSD measurements from Voigt et al. (2017) and Atlas et al. (2005) have been added to Figure 7.
l. 593: "*PSD measurements in natural cirrus and aged contrails differ significantly depending on the location of the study, ambient air conditions, measurement methods (instrument limitation (Gayet et al., 2002), and air speed (Febvre et al., 2009)). Previous studies show that a 3-hours old contrail cirrus with an effective diameter close to 20 µm (Voigt et al., 2017) and number*

*concentration larger than 0.1cm⁻³ (Schumann et al., 2017) can be composed of ice crystals with sizes up to 100 μm (blue dashed line, contrail cirrus figure 7). This differs from the PSD of the natural cirrus presented by Voigt et al. (2017) (dashed black line), which has an order of magnitude lower particle number concentration. In natural cirrus at mid-latitudes, ice crystals with size up to 1600 μm were observed during the ML-CIRRUS campaign (dark dashed line Figure 7, Voigt et al., 2017).*"

20) Table 2: (a) I suggest to sort the clusters like recommended under Point 16 f). A further suggestion is to sort Ntotal in two size intervals, namely <~30um and >~30um, since the grouping of the clusters change with size.
(b) The mean and median values of IWC does not fit to each other. For example, for PW / CC the means are 15.46/28.69 mg/m3, but the medians are 6.26/0.96 mg/m3, i.e. the mean of CC is almost twice the mean of PW, but the median of CC is much lower than that of PW ?
Please check all numbers.
(c) Mean/median of Ntotal for CC are 6.06/3.75 cm -3. This is too high for natural cirrus – from Voigt et al. (2017), I would expect something around 0.1 cm -3 or even lower. Is that an arithmetical error , shattering or could it be that contrails are accidentally attributed to CC ? Please clarify!

(a). Table 3 has been sorted according to the development of contrails as proposed by the reviewer in point 16.f).

(b). The new definition of the clusters does not allow to fit mean and median values. This observation may be due to the hypothesis used for PSD definition such as particle sphericity and the interpolation realized between 17 μm and 70 μm.

(c). As mentioned in the previous point, PSD of natural cirrus are significantly different according to measurement location and the different probes used. Here, the new clustering method shows lower number concentrations for the "natural cirrus".

Modifications:
l.623: "*Despite the large uncertainties associated to both instruments and the interpolation between 17 μm and 70 μm diameters, these results again show that each cluster can be connected to a specific contrail phase, and their properties can be compared to previous studies.*"

21) One last comment: could you discuss the possibility to use other/more parameters for the PCA? For example, could Ntotal be included in the PCA ? Or in case no Polar Nephelometer is on board, but PSD, IWC and NO is available, do you think the analysis would be possible?

The present paper demonstrates that the PCA method allows contrail classification from optical properties only. Indeed, additional parameter could improve the performance of the PCA method as PSD measurements, RHI values and NO concentrations. However, the additional parameter should be carefully selected in order to limit the bias introduced by the limitations of the probes and can vary from a measurement campaign to another.
It has been mentioned in the text.

Modifications:
l. 688: "*The additional use of microphysical and chemical measurements can be added to the PCA method in order to improve the selection of contrail phases. Different ranges of extinction or asymmetric coefficients could be also used for PCA analyses in this perspective. However, additional parameters should be carefully selected to limit the bias introduced by the limitations of the probes and the optimal selection may vary from one measurement campaign to another.*"

RC2
The authors present aircraft observations of the scattering properties of ice crystals and the trace gas properties sampled inside 17 contrails during two phases of the CONCERT field experiment. While the results presented here are relevant and interesting, the paper has several areas where more explanation is warranted before I can recommend it for publication. For example, some parts of the introduction need to be reorganized.

RC : The most major flaw of the paper which needs to be address is the selection of the clusters. The authors base their cluster classification on a rough examination of the first three principal components in the x-y plane and seem to draw ellipses around where they "roughly identify" where the clusters are. However, with recent advances in machine learning, there are more objective methodologies for classifying data into clusters, with the most applicable methodology for a feature space of three variables to be k-means clustering. The authors should either better justify why their current ellipses were chosen and why the feature space was used for the PCA, or use automated clustering techniques.

Finally, I think a section on how their contrail cirrus observations fit in with past studies is warranted, since the paper lacks much discussion on how their observations fit in with what is already in the literature. I list some other comments below.

We would like to thank the reviewer for his interesting and constructive suggestions. We have tried to follow every suggestion in order to improve the manuscript. Each reviewer's comments are addressed and the manuscript has been modified accordingly.

A significant change in the paper concerns the implementation of an automatic clustering method (k-mean method) to enhance the statistical significance of the contrail phase discrimination. Subsection "*3,2,2 Clustering analyses*" has been added and gives details of the method. After some tests we found that 16 clusters are necessary to classify the patterns revealed by the PCA analysis. To be in accordance with ATC observations, clusters 8 to 16 deduced from the k-mean method (Figure RC2.2.a.) were gathered into two clusters (A and B, Figure RC2.2.b.). Clusters 2 to 5 have also been merged to one single cluster.

[Figure]

Figure RC2.1.: Clustering analyses according k-mean method. a) First step using 16 clusters and b) second step grouping clusters 8 to 16 to two clusters (A and B).

The following table summarizes the correspondence of the clusters defined by the k-mean methods and the cluster's definitions according to ATC information:

| k-mean clusters | Cluster number | definition | name |
|---|---|---|---|
| 1 | 0 | Primary Wake | **PW** |
| 6 | 1 | Young Contrail 1 | **YC1** |
| 2, 3, 4 and 5 | 2 | Young Contrail 2 | **YC2** |
| A | 3 | Aged Contrail 1 | **AC1** |
| 7 | 4 | Aged Contrail 2 | **AC2** |
| B | 5 | Cirrus Cloud | **CC** |

Table RC2.1: Cluster definition according the k-mean method.

Comments:

Major comments:
Lines 51-91. This paragraph is too long and needs to be reorganized. For example, there is too much detail on how NO from aircraft exhaust is converted into acids that does not really add to the major point that NO interacts with OH to make nitr(ic)ous + sulfuric acid. I also feel that this can really be 3 paragraphs: one about NO interacting with OH to produce acids, one about the contrail production process and one about the contrail aging process.

The structure of the introduction has been changed and two subsections were added, namely:
        1.1. Contrail formation and evolution
        1.2. Optical and microphysical properties of contrail phases

Moreover, the description of contrail chemical properties has been shortened.

Modifications:
l.55: "*Several studies in the past have been dedicated to the evolution of concentrations of nitrogen oxide (NO) and sulphur dioxide ($SO_2$) and their oxidized forms (Kärcher and Voigt, 2006 ; Voigt et al., 2006 ; Schäuble et al., 2009 ; Jurkat et al., 2011).*"

Line 109-146: I feel that a lot of the individual data points cited here are better suited for an extra section in the paper comparing your contrail observations against past studies. Right now, no link is made to how your categories compare against these past observations and I think such a comparison is needed in order to justify that the range of values that you observe in your clusters correspond to contrails properties that are observed in nature. Therefore, I recommend shortening this paragraph to
just briefly explain how the microphysical properties of contrails evolve with time with leaving specific numbers to a later comparison.

The paragraph has been shortened and we compared our contrail observations against past studies in section 4.2. In particular, average microphysical properties of the clusters were compared with the main findings of Voigt et al. (2017), Schumann et al. (2017) and Atlas et al. (2005). We also discussed the shortcomings related to the interpolation of the PSD and its impact on the derived microphysical quantities.

Modifications:

[Figure]

Figure 7: Number particle size distributions for each cluster including all data points of all flights. FSSP-300 measurements from 0.5 to 17 µm and 2DC measurements from 70 µm to 800 µm. The data are linearly interpolated in logarithm space between 17 µm and 70µm.

Voigt et al. (2017) and Atlas et al. (2005) PSD measurements have been added to Figure 7 to get a better picture of previous results. It should strengthen the statements on contrail properties discussed in this section.

l.591:"*Because of this gap, the derived microphysical properties should be considered with caution, but may be used to check the cluster definitions.*"

l. 595: "*Previous studies show that a 3-hours old contrail cirrus with an effective diameter close to 20 µm (Voigt et al., 2017) and number concentration larger than 0.1cm$^{-3}$ (Schumann et al., 2017) can be composed of ice crystals with sizes up to 100 µm (blue dashed line, contrail cirrus figure 7). This differs from the PSD of the natural cirrus presented by Voigt et al. (2017) (dashed black line), which has an order of magnitude lower particle number concentration. In natural cirrus at mid-latitudes, ice crystals with size up to 1600 µm were observed during the ML-CIRRUS campaign (dark dashed line Figure 7, Voigt et al., 2017).*"

l.630: "*These properties are in agreement with previous measurement reported by Gayet et al. (2012) with particle number concentrations close to 200 cm$^{-3}$ for contrails less than 60 s after their formation. Their work also reports extinction coefficient around 7 km$^{-1}$ presenting the highest values of the contrail life time.*"

l.638: "*The ice number concentrations are in agreement with previous results with values between 200 and 100 cm$^{-3}$ for contrail ages between 60 s and 3 min, and around 5 cm$^{-3}$ for contrail ages around 10 min (Goodman et al., 1998 ; Lawson et al., 1998 ; Schröder et al., 2000 ; Schäuble et al., 2009 ; Gayet et al., 2012 ; Voigt et al., 2017).*"

l.645: "*However, the ice number concentration and the extinction coefficient are higher than in previous studies, with values around 0.1 cm$^{-3}$ and 0.023 km$^{-1}$ respectively.*"

Line 223-225: You aren't using the 2DC for calculating IWC though! I don't see why this sentence is needed. However, I think text here justifying why you are not using observations below 70 microns due to the 2DC's limited response time and depth of field need to be here.

Optical and microphysical properties of contrail particles cannot be fully retrieved without the 2DC measurements. In the present study, these properties are retrieved with both instruments FSSP and 2DC, to consider as much as possible the full size range of the particle size distribution. A linear interpolation has been applied between the two instrument ranges. The text has been clarified according to this remark.

Modification:
l.191: "*Particle size distributions and corresponding microphysical and optical integrated properties (IWC, Deff, N, and extinction) were derived from both FSSP-300 and 2DC measurements.*"

Section 3.2: I think more justification needs to be given for the choice of your feature space for the PCA, since right now it is presented without really linking the feature space to looking for quantities that we expect to vary in differing stages of contrail cirrus. For example, why did you conduct a PCA on the entire scattering phase function instead of just apply clustering to the asymmetry parameter?

Also, why were the clusters manually chosen instead of using automated techniques like k-means clustering?

We thank the referee for this very interesting point and for the new perspective brought to the interpretation of our results. The k-mean clustering method was applied to our dataset. We found that this clustering method lead to an accurate (and more robust) classification of every contrail phase and natural cirrus which agrees with ATC information.
The asymmetry parameter information is not sufficient to define the different clusters (Jourdan et al., 2010). Indeed, gPN information cannot be used to separate the group YC1/YC1 and AC1/AC2/CC, which represent the most interesting part of the study. Additionally, Jourdan et al. 2010 and 2003 showed that the asymmetry parameter alone cannot represent or mimic the variability of the phase function. Indeed, the information content of a phase function measurement can be used to discriminate ice clouds characterized by different ice crystal shape, size or degree of surface roughness which is not the case with the g factor alone. Our study here clearly shows that the first 2 principal components (which are correlated to the extinction coefficient and the g factor respectively) cannot reproduce the whole variability of the optical (and microphysical) properties. The third principal component adds additional valuable information on contrail type which is not directly related to the asymmetry parameter.

Modifications:
The k-mean method has been applied to the dataset and an additional subsection has been added ("*3,2,2 Clustering analyses*). The new clusters are thus defined as mentioned above.

Lines 505-512: How do you know that you flew in an aged contrail with no verification from ATC? I think the important conclusion here is more that, microphysically, aged contrails and cirrus are very similar and are difficult to distinguish with this data alone.

For flight 17 of CONCERT 2, ATC reports chasing but without any possibility to check which aircraft is actually followed. This information indicates that measurements may correspond to very aged contrails or natural cirrus. From in-situ measurements, aged contrails and natural cirrus are very similar. In our case, the PCA method along with K-mean clustering method classifies the 17 C-2 measurement in both aged contrail and natural cirrus (AC1 and CC) with both a significant number of point. This can be explained by a limitation of the method to separate aged contrail and natural cirrus due to quite similar optical properties, but also by a mixture of both natural cirrus and aged contrails. Indeed, differences exist in the [60°-80°] range as well as in the forward and backward scattering regions. These differences can be significant as they are detected by the PCA and the clustering method.

Modifications:
l.505: "*Still ATC data indicate measurements in exhaust plumes and the Falcon flew apparently in visible contrails (ExtPN > 0.1 km$^{-1}$) which were probably too old for ATC recognition.*"

Line 518-522: I think this analysis can be better supported by showing the distributions of contrail ages from ATC.

Only an estimation of contrail age range is available. We couldn't derive a precise age for each individual contrail sampled during each flights. Table 2 shows the available age ranges for the identified contrails.

Line 593-595: I would not interpolate data in this range since the interpretation of extrapolated data could be quite dangerous. I would simply state that concentrations in this size range are too uncertain to report due to the 2DC's poorly characterized depth of field and response time.

We agree with this comment as the interpolation between the two instruments can induce large uncertainties when calculating the microphysical properties. However, we choose to keep this approximation in order to retrieve microphysical properties comparable to previous studies.

Modifications:
l.623: "*Despite the large uncertainties associated to both instruments and the interpolation between 17 μm and 70 μm diameters, these results again show that each cluster can be connected to a specific contrail phase, and their properties can be compared to previous studies.*"

Lines 607-610: Your YC1 contrails seem to have roughly similar 2DC number concentrations to the aged contrails. Why is that?

The new clustering method refined the cluster definition. Consequently, it leads to a better partitioning of contrail microphysical properties. Indeed, as shown by the new Figure 7, the mean PSD from cluster YC1 displays significantly less particle in the 2DC measurement size range than the one corresponding to AC1 and AC2 clusters.
As already discussed into the text, differences between each mean PSD should be taken carefully due to uncertainties of both probes.

Modifications:
These discussion has been added into the text l.616:"*Within the 2DC range, the PSDs are also in agreement with the cluster definitions. A higher concentration of large ice crystals with diameters around 100 μm and larger are expected for natural cirrus (cluster CC) and for significantly well-developed contrails. This is particularly well illustrated by the mean PSD from cluster YC1 that*

*displays significantly less particles in the 2DC measurements size range than the one corresponding to AC1 and AC2."*

Lines 640-667: I would convert this into a bulleted list of conclusions to make this paragraph easier to read.

A summary of contrail property is more clearly presented in two separate paragraphs on the conclusion.

Figures/Tables:

Figures 5c,d: A logarithmic x-axis would make the lines easier to distinguish.
Figure 6: I would advise removing the lines where you don't have the PSD from the two probes in the ~20 to 70 micron range. Can you also add size distributions from past studies and include them in the comparison?

Table 2: I think some data from contrails sampled in past studies should be shown and compared against here and in the paragraph discussing Table 2.

Figure 6 c,d : The suggested modifications have been taken into account.

Figure 7: Previous PSDs (Voigt et al., 2017 and Atlas et al., 2005) have been added to the Figure as reported in previous comments of the reviewer. However, we believe that interpolation illustrations at this stage of the paper is essential in order to understand microphysical properties retrieved from this approximation.

Table 3: Results from past studies are now discussed in the text.

[revised manuscript text omitted]

---

## Editor Decision (ED1)

**Reviewer Comments**

In my original review I wrote:

*.... my overall rating is major revisions because of two points emphasized here:*
*– ....*
*– Some numbers in Table 2 needs to be checked, the mean and median IWC and,*
*especially, the mean and median of Ntotal for CC are too high for natural cirrus – see*
*comment 20).* This number is definitely not ok, obviously contrails are mixed in the
natural cirrus class.

Here is the specific point of my review:

**Point 20 c)**
**Mean/median of Ntotal for CC are 6.06/3.75 cm-3. This is too high for natural**
**cirrus. From Voigt et al. (2017), I would expect something around 0.1 cm-3 or even**
**lower.**

And here is the answer to it:

*PSD of natural cirrus are significantly different according to measurement location and*
*the different probes used. Here, the new clustering method shows lower number*
*concentrations for the "natural cirrus".*

In the new manuscript one finds:

Mean/median of Ntotal for CC are 5.092 / 3.444  cm-3

which is nearly the same as before (and not lower !!) -  and it is still much too high (see
the plot below), so the authors didn't take this major comment  seriously.

Wrt the argument that N_ice greatly vary with measurement location:  yes,
but  observations> 1 cm-3 are exceptions, and unrealistic as mean or median values at
any location....

In the middle plot (Voigt et al. 2017, ML-Cirrus, 18 hours of  N_ice observations)  you
can see that already a value of 1 cm-3 is rarely exceeded. During ML-Cirrus  lots of
contrails  were observed,  representing almost all higher values in the Figure, in natural
cirrus the frequency of cirrus with N_ice > 1 cm-3 is much smaller. Voigt et al. (2017):

In Table 3, the median of Ntotal of AC1 (Aged Contrail 1) is 1,696 cm-3 , while in CC it is 3,444 cm-3. Also, the 25% and 75% percentiles are lower for AC1 than for CC. How can mean/median the ice particle concentrations be lower in aged contrails than in natural cirrus ?

If the mean/median ice particle numbers in CC in Table 3 are not typos (what I thought when I first read the paper), but are now 5.092 / 3.444 cm-3, then either the method is called into question (that was the reason that I rated this point as major) or the data base is too small.

How large is the data base, and how much sampling time is spend in the different classes ?

[Figure]

FIG. 6. (a) Range and temperature dependence of the IWC detected during ML-CIRRUS derived from HAI/SHARC hygrometers (blue dots) and median from Schiller et al. (2008) (black line). (b) Ice number densities in the size range of 3- to 937-μm diameter ($N_i$) in cirrus from NIXE–CAPS and middle and maximum $N_i$ from Krämer et al. (2009). (c) Relative frequency of RHi in cirrus from AIMS–H2O (Kaufmann et al. 2016) and Basic Halo Measurement and Sensor System (BAHAMAS) temperature data in 1-K temperature bins. The light gray line shows the homogeneous nucleation threshold from Koop et al. (2000) and the dark gray line shows the liquid water saturation (Murphy and Koop 2005).

---

## Author Response (AR2)

**Reviewer Comments**

In my original review I wrote:

*.... my overall rating is major revisions because of two points emphasized here:*

*– ....*

*– Some numbers in Table 2 needs to be checked, the mean and median IWC and, especially, the mean and median of Ntotal for CC are too high for natural cirrus – see comment 20). This number is definitely not ok, obviously contrails are mixed in the natural cirrus class.*

Here is the specific point of my review:

**Point 20 c)**

**Mean/median of Ntotal for CC are 6.06/3.75 cm-3. This is too high for natural cirrus. From Voigt et al. (2017), I would expect something around 0.1 cm-3 or even lower.**

And here is the answer to it:

PSD of natural cirrus are significantly different according to measurement location and the different probes used. Here, the new clustering method shows lower number concentrations for the "natural cirrus".

In the new manuscript one finds:

Mean/median of Ntotal for CC are 5.092 / 3.444  cm-3

which is nearly the same as before (and not lower !!) -  and it is still much too high (see the plot below), so the authors didn't take this major comment  seriously.

Wrt the argument that N_ice greatly vary with measurement location:  yes, but  observations> 1 cm-3 are exceptions, and unrealistic as mean or median values at any location....

In the middle plot (Voigt et al. 2017, ML-Cirrus, 18 hours of  N_ice observations)  you can see that already a value of 1 cm-3 is rarely exceeded. During ML-Cirrus  lots of contrails  were observed, representing almost all higher values in the Figure, in natural cirrus the frequency of cirrus with N_ice > 1 cm-3 is much smaller. Voigt et al. (2017):

In Table 3,  the median of Ntotal  of AC1 (Aged Contrail 1) is  1,696 cm-3 , while in CC it is   3,444 cm-3. Also, the 25% and 75% percentiles are lower   for  AC1 than for CC. How can mean/median  the ice particle concentrations be lower in aged contrails than in natural cirrus ?

If the mean/median  ice particle numbers in CC in Table 3 are not typos (what I thought when I first read the paper), but are now 5.092 / 3.444 cm-3, then either the method is called into question (that was  the reason that I rated this point as major) or the data base is too small.

How large is the data base, and how much sampling time is spend in the different  classes ?

[Figure]

Fɪɢ. 6. (a) Range and temperature dependence of the IWC detected during ML-CIRRUS derived from HAI/SHARC hygrometers (blue dots) and median from Schiller et al. (2008) (black line). (b) Ice number densities in the size range of 3- to 937-$\mu$m diameter ($N_i$) in cirrus from NIXE–CAPS and middle and maximum $N_i$ from Krämer et al. (2009). (c) Relative frequency of RHi in cirrus from AIMS–H2O (Kaufmann et al. 2016) and Basic Halo Measurement and Sensor System (BAHAMAS) temperature data in 1-K temperature bins. The light gray line shows the homogeneous nucleation threshold from Koop et al. (2000) and the dark gray line shows the liquid water saturation (Murphy and Koop 2005).

**Answer from authors:**

We apologise sincerely to the reviewer for these inaccuracies. We understand his questioning on the microphysical properties of natural cirrus (cluster CC) and will try with this answer to give more details on the previous explanation (l. 655) of the high concentration numbers found in the natural cirrus cluster:

L655: "*However, the ice number concentration and the extinction coefficient are higher than in previous studies, with values around 0.1 cm-3 and 0.023 km- 1 respectively. Besides to interpolation between the FSSP-300 and the 2DC measurements, the assumed shape (spherical or aspherical), and shattering of large ice particles in cirrus and aged contrails can also have a significant effect on the measurement of optical and microphysical properties (Gayet et al., 2012).*"

First, as mentioned previously in the manuscript, the accuracies on microphysical parameters derived from both FSSP-300 and 2D-C probes are seriously hampered by the combination of inherent limitations of these two old probe versions. The effects of shattering of large ice particles (typically larger than 100 $\mu$m) on FSSP probe tips may significantly increase the number concentration of small particles (see among others Febvre et al., 2009) whereas the use of the 2D-C at Falcon airspeed (~ 180 m s$^{-1}$) lead to large sizing and counting errors (for size lower than 100 $\mu$m) due to optical and electronic limitations (Lawson et al., 2006).

The Particle size distribution of Cluster CC now called PC (polluted cirrus, in blue) on figure 7 shows high concentrations of ice crystals in the 2D-C size range (from 50 to 800 $\mu$m). Shattering of large ice crystals on the FSSP and 2DC probe tips may result in a significant increase of the concentration of small particles in the FSSP size range and a decrease of the number concentration in the 2D-C size range.  Indeed, our results show that high particle concentrations (even higher than AC1 cluster) are found in the FSSP size range (between 0,5 and 18 µm) which is not in agreement with previous studies on cirrus microphysical properties (Voigt et al., 2017 for instance).

New instruments (such as CDP, 2D-S, CAS …) can now be equipped with new tips designed to greatly reduce the production of ice fragments from shattering (Korolev et al., 2011). Software techniques detecting closely spaced particles (inter arrival time) assumed to result from shattering are also available to account for shattering contamination of the measurements. Unfortunately, our measurements were based on "old" instruments for which these techniques were not (or couldn't be) applied. Whereas the shattering of large ice crystals should not be significant in contrails, it could have strongly impacted our measurements in cirrus clouds (CC cluster). This effect is now discussed thoroughly in the text and number concentration values for ice crystals with size larger than 50 µm are also mentioned.

Secondly, as mentioned in the manuscript, the interpolation technique used to derive PSD in the 17 µm – 50 µm size range, not accurately measured by the FSSP nor the 2DC, can also lead to large uncertainties in the derived number concentration.  Voigt al., 2017 showed that particles in this size range account for one third of the total concentration measured in natural cirrus and contrail cirrus (see figure 7).

l. 704: "*A linear interpolation in logarithmic space is applied for each PSD in the size range from 17 µm to 50 µm to cover the gap from 17 µm to 50 µm. Because of this gap, the derived microphysical properties should be considered with caution, but may be used to check the cluster definitions.*"

Our results still show that the PCA combined with a K-means classification can be used to clearly discriminate young from aged contrails based on their scattering properties only. This method is thus an interesting approach to analyse the contrail evolution. However, with this dataset, the method cannot clearly separate the scattering properties of aged contrails from the natural cirrus properties. Cluster CC presents significant NO concentrations with microphysical properties close to aged contrails, and we cannot exclude that the higher ice number densities in the cirrus compared to the mean in Voigt et al., 2017 might be explained by an aviation impact. Therefore, the term "natural cirrus" should be changed to "polluted cirrus". This has been modified in the text after having discussed the results in Figure 6. Indeed, the cirrus observed during CONCERT 1 and 2 could have been influenced by aviation with high traffic density over Germany.

To answer the last question of the reviewer, the number of used cases per cluster for the microphysical property calculations (need to have both complete FSSP and 2DC measurements), are:

- Cluster PW: 482 points
- Cluster YC1: 141 points
- Cluster YC2: 2369 points
- Cluster AC1: 6833 points
- Cluster AC2: 470 points
- Cluster CC: 1655 points

The corresponding times spent on each cluster are difficult to retrieve due to the mixing of data from 8 flights. However, 1 point corresponds to 1 second of measurements. Thus, cluster CC and AC1 have a sufficient number of data points to derive mean values for the measured properties in both clusters.

**Modifications:**

In the all text, "natural cirrus" has been replaced by "cirrus" (cluster CC). Then, NO studies allow characterizing "cirrus" as "polluted cirrus" (cluster PC). Because two different clusters can be defined between "aged contrail" and "polluted cirrus" from the k-mean clustering method, these two clusters are still studied separately.

l. 229: "*These equations do not account for possible shattering of large ice crystals on the probe inlets. This effect is minor in young contrails but can lead to an underestimation of large ice crystal concentration (diameters higher than 100 μm) and thus an overestimation of small ice crystal concentration in contrail cirrus clouds (Febvre et al., 2009).*"

l. 565: "*Due to the high and similar nitrogen oxide concentrations in clusters AC1 and CC, we can conclude that the clouds initially classified as "natural cirrus" are, in fact, significantly influenced by high-density air traffic over Germany. In what follows, these parts of CONCERT measurements are classified as "polluted cirrus" (cluster PC).*"

l. 605: "*In addition, the differences observed between the PSD of PW/YC1/YC2 and AC1/AC2/PC can be explained by the production of small ice crystals (from 1 to 10 μm) in fresh exhaust plumes followed by rapid dilution during subsequent minutes after the exhaust. It is important to note that aged contrail measurements classified into the AC1 cluster present significantly lower ice particle concentrations than polluted cirrus. The small differences between the two clusters in optical and chemical properties may be explained by strong shattering effects, as mentioned previously. Indeed, the shattering of large ice particles (diameters larger than 100 μm) can increase the particle number concentrations significantly (Febvre et al., 2009).*"

l. 613: "*Even if shattering effects influence ice particle concentrations within the 2DC range, the PSDs are still consistent with the cluster definitions.*"

l. 642: "*Cluster PC corresponds to polluted cirrus. The IWC is significantly higher (28 mg m$^{-3}$) within this cluster than in other clusters, and higher than observed in previous studies for clean natural cirrus. Also the ice number concentration and the extinction coefficient for cluster PC are higher than for clean cirrus, with values around 0.1 cm$^{-3}$ and 0.023 km$^{-1}$ respectively. As mentioned in section 4.1, cirrus observed during CONCERT campaigns are largely influenced by high-density air traffic over Germany and it is thus still difficult to separate aged-contrails and natural cirrus based on their scattering properties. In addition, shattering effects may have significantly influenced the number concentrations of ice particle as discussed previously (section 2.2). Indeed, if only particles with diameters larger than 50 μm are analysed, which better corresponds to an expected cirrus range, the mean number concentration for the polluted cirrus cluster is 0.001 cm$^{-3}$.*"

l. 675: "*However, no strictly clean cirrus has been observed during these two campaigns due to a too strong influence from dense air traffic over Germany.*"

[revised manuscript text omitted]

---

## Author Response (AR3)

Answer to the author's reply of May 8:

It would be really a pity if this nice study would suffer from ice particle measurements contaminated by shattering. And I'm not convinced whether the high ice concentrations in natural cirrus are caused by shattered ice fragments. First, the observed ice particles are not large enough to cause shattering (see Fig. 10 of Voigt et al., 2010, ACP). Their maximum sizes are below 500 µm diameter, while shattering starts to occur in the presence of larger ice particles. Second, in cases of shattering the concentrations are mostly much higher than those shown here.

The second argument presented by the authors to explain the enhanced ice concentrations are uncertainties due to the interpolation of the PSD in size range of 17 µm – 50 µm. To my opinion, this is also unlikely since the concentrations in this size range are not high enough.

I propose another idea where the higher concentrations could come from: from a closer look into Voigt et al. (2010) I got the impression that the ice particle concentrations are derived by integration over the total FSSP size interval of 0.45 µm to 17.7 µm instead of considering only the cloud particles > 3 µm (see Voigt et al., 2017). That would mean that aerosol particles are added to the ice concentrations, which would explain the bias in the data.

This could be easily checked and if it is the case, non-contaminated data could be presented in the paper, which would make it scientifically more sound.

To publish data including shattering seems problematic to me, though I understand the point that the applicability of the PCA method can be demonstrated on the basis of the current data set.

We thank the reviewer for taking in charge the review of the present paper and allowing minor revisions for the publication. The manuscript has been further improved as a consequence of the suggestions of the reviewer, both on the cluster definition and the calculation of microphysical properties.

First, in order to be clear about ice cloud definitions from CONCERT's measurements, the cloud event which we called "natural cirrus" before is now called "unidentified ice cloud". Indeed, we have no ATC information to classify this event as contrail, but extinction and asymmetric coefficients show that these measurements occurred in a significantly thick ice cloud. Due to pollution by intense air traffic in this region, the cluster defining this part of the measurements is then called "polluted-cirrus" (cluster PC).

**Modifications:**

The term "natural-cirrus" has been replaced by "unidentified ice cloud" in the all relevant text parts and related tables/figures, and the term "polluted cirrus" or "PC" is used for cluster 5.

l. 267: "*When no ATC information is available, the cloud segment is called "unidentified ice cloud".*"

l. 279: "*The last cloud event ("unidentified ice cloud") during flight 16b is not a contrail because it is measured at temperatures significantly above the Schmidt Appleman temperature (-38°C, Schumann 1996). This is an ice cloud with high extinction (> 0.5 km$^{-1}$) and low asymmetry values (<0.75), characteristic for ice particles (Jourdan et al., 2003b, Febvre et al., 2009). Relative humidity and NO mixing ratio data are not available for this cloud.*"

l. 435: "*9 of the original clusters are merged into 2 clusters (clusters 3 and 5) presenting similar NO concentrations and optical properties.*"

l. 449: "According to *ATC information, these clusters both contain parts of the measurements in the B767, A343, A346 and CRJ-2 contrails. In addition, the unidentified ice cloud event from flight 16b is fully included in cluster 5. Unpolluted natural cirrus was rarely observed during the CONCERT campaigns (Voigt et al., 2010). Since we have no objective way of discriminating natural cirrus from contrail cirrus region, these clouds are referred to polluted cirrus or PC, and cluster 3 to aged contrails.*"

L. 699: "*The optical and microphysical properties of the aged contrails are often similar to those found in ambient cirrus which may be polluted cirrus.*"

Secondly, the number concentrations were calculated from the full diameter range of the FSSP and the 2DC. It means that the range from 0.5 to 800 µm was considered.

In Voigt et al. (2010) particle size distributions were derived from FSSP and 2DC considering the full size range. It was discussed that cirrus cases may have been affected by particle shattering because large ice crystals were detected by the 2DC. The contrail FSSP-300 measurements seemed not to be strongly affected by ice shattering since the cirrus contribution to contrail ice crystal surface or volume distribution for particles smaller than 17.7 µm was less than 1% (figure 10 Voigt et al., 2010).

Voigt et al. (2017) chose to select particle diameters higher than 3 µm to retrieve the number concentration of cirrus ice particles. In addition, we know that contrail cirrus occurs at ambient temperatures below -38°C.

Hence, the extinction coefficients, the Ice Water Content, and the number concentration for aged contrail clusters and the polluted cirrus cluster have been calculated for temperatures lower than -38°C. In addition, for the same clusters, optical and microphysical properties have been calculated for diameters higher than 3 µm. Results are in better agreement with previous results including those shown by Voigt et a. (2017) and in other studies of natural cirrus. We choose to present both concentrations (for the complete size range and for diameters larger than 3 µm) for comparison purposes.

**Modifications:**

Figure 7: Particle size distributions of aged contrails and polluted cirrus clusters have been modified according the temperature threshold of -38°C.

[Figure]

Figure 7: Number particle size distribution for each cluster including all data points of all flights. FSSP-300 measurements from 0.5 to 17 μm and 2DC measurements from 50 μm to 800 μm. The data are linearly interpolated in logarithm space in the gap between 17 μm and 50 μm.

l. 610: "*Higher concentrations of ice crystals with diameters larger than 100 μm are observed for polluted cirrus (cluster PC) and for well-developed contrails (cluster AC1). The average PSD of AC1 cluster shows much larger ice concentrations (around 10 times) compared to YC1 cluster within the 2DC size range.*"

l. 614: "*It is important to note that shattering effects can significantly influence the PSD measurements especially when particles with diameters higher than 100 μm are present. Polluted cirrus or aged contrail measurements could be subject to such artefacts even though the concentrations of large ice particles were low in the aged contrails and in the polluted cirrus cases during these two campaigns. Shattering effects are likely to be small for the measurements in young contrails.*"

Table 3: Values for ambient temperatures higher than -38°C have been included in the table, and values for particles with diameters higher than 3 μm have been added to the table for aged contrails and the polluted cirrus clusters (AC1, AC2 and PC). The values for the limited size range are given in parenthesis.

| Extinction (km⁻¹) | | mean | | std | Median | prctile 25 | prctile 75 |
|---|---|---|---|---|---|---|---|
| cluster | **PW** | 4.230 | | 3.820 | 3.308 | 1.104 | 6.485 |
| | **YC1** | 0.720 | | 0.410 | 0.680 | 0.351 | 1.026 |
| | **YC2** | 2.070 | | 2.655 | 1.017 | 0.271 | 2.836 |
| | **AC1** | 0.212 | (0.204) | 0.465 (0.456) | 0.037 (0.033) | 0.008 (0.005) | 0.152 (0.138) |
| | **AC2** | 0.114 | (0.090) | 0.163 (0.149) | 0.060 (0.038) | 0.007 (0.003) | 0.135 (0.094) |
| | **PC** | 0.207 | (0.197) | 0.363 (0.360) | 0.072 (0.062) | 0.032 (0.026) | 0.178 (0.160) |

| IWC (mg m⁻³) | | mean | | std | Median | prctile 25 | prctile 75 |
|---|---|---|---|---|---|---|---|
| cluster | **PW** | 8.173 | | 10.586 | 5.573 | 1.665 | 11.363 |
| | **YC1** | 0.191 | | 0.107 | 0.168 | 0.111 | 0.281 |
| | **YC2** | 4.860 | | 8.918 | 1.235 | 0.218 | 6.604 |
| | **AC1** | 5.707 | (5.705) | 25.120 (25.120) | 0.124 (0.122) | 0.007 (0.004) | 1.126 (1.123) |
| | **AC2** | 0.310 | (0.304) | 1.103 (1.103) | 0.112 (0.093) | 0.005 (0.002) | 0.290 (0.285) |
| | **PC** | 3.024 | (3.022) | 8.845 (8.845) | 0.218 (0.214) | 0.080 (0.079) | 0.641 (0.639) |

| NTOTAL (cm⁻³) | | mean | | std | Median | prctile 25 | prctile 75 |
|---|---|---|---|---|---|---|---|
| cluster | **PW** | 172.965 | | 114.497 | 152.398 | 95.564 | 223.374 |
| | **YC1** | 409.726 | | 205.625 | 405.127 | 230.907 | 603.187 |
| | **YC2** | 188.139 | | 199.736 | 125.344 | 52.584 | 236.100 |
| | **AC1** | 8.148 | (0.372) | 24.646 (2.103) | 1.688 (0.086) | 0.027 (0.027) | 3.311 (0.179) |
| | **AC2** | 29.517 | (0.427) | 44.723 (1.005) | 8.021 (0.128) | 0.0120 (0.020) | 46.762 (0.290) |
| | **PC** | 6.646 | (0.360) | 7.237 (0.864) | 4.602 (0.213) | 0.110 (0.110) | 8.354 (0.394) |

Table 3: Optical and microphysical properties for each cluster according interpolated particle size distributions from FSSP-300 and 2DC measurements. Values in parenthesis correspond to number concentrations for sizes larger than 3 µm.

l. 622: "*The aged contrail clusters (AC1 and AC2) and the polluted cirrus cluster (PC), include some data points at temperatures higher than -38°C. These values cannot be contrails and are excluded from this analysis. Ice particles with diameters higher than 3 µm are considered for aged contrails and polluted cirrus to exclude possible contributions from large aerosol particles, as in the earlier studies of Krämer et al. (2009) and Voigt et al. (2017), and these values are shown in parenthesis. These results again show that each cluster can be related to a specific contrail phase, and their properties can be compared to previous studies.*"

l. 640: "*Indeed, the averaged extinction and number concentration values of aged contrails do not exceed 0.4 km⁻¹ and 30 cm⁻³ (0.5 cm⁻³ for diameters higher than 3 µm), respectively.*"

l. 645: "*For aged contrail, concentrations of ice particles with sizes greater than 3 µm are below 0.5 cm⁻³, which is in agreement with concentrations presented in other contrail studies. Also IWC values and extinction coefficients of aged contrails agree with previous studies and their values are only weakly sensitive to the cut-off size used (below or above 3 µm).*"

l. 653: "*The polluted cirrus IWC is significantly higher (3.02 mg m⁻³) than observed in previous studies for clean natural cirrus. The same holds for ice number concentration and extinction coefficient with values of 6.66 cm⁻³ and 0.21 km⁻¹.*"

l. 659: "*When limiting the analysis to ice particles larger than 3 µm, the ice number concentrations for the polluted cirrus have mean values of 0.36 cm⁻³, which is in better agreement with previous cirrus studies. However, IWC and extinction coefficients values (3.02 mg m⁻³ and 0.20 km⁻¹, respectively) are still significantly higher than for clean cirrus cases observed in previous studies (0.001 mg m⁻³ and 0.023 km⁻¹, respectively). Their optical and microphysical properties are closer to aged contrail*"

*properties. This is consistent with our interpretation that high air traffic emissions in the measurement region may have influenced the cirrus collected in cluster PC."*

l. 700: "*For polluted cirrus, the agreement with previous cirrus data is better when considering only ice particles with diameter higher than 3 μm.*"

Finally, some editorial improvements have been made to improve English language and clarity of the paper.

---

## Author Response (AR4)

**Comments to the Author:**

-- on page 22 'previous studies' are mentioned several times without a reference.

-- also on page 22 it is stated that 'the polluted cirrus IWC is significantly higher (3.02 mg m -3) than observed in previous studies for clean natural cirrus.' (no reference).

But, comparing mean/median IWC presented here with those reported by Schiller et al. (2008) and Krämer et al. (2016) show a good agreement.

Later in the text a mean IWC of 0.001mg/m3 from previous studies is mentioned for clean cirrus. Where does it comes from? This is definitely at the low end of observed natural cirrus IWC.

-- also page 22: 'The same (sigificantly higher values) holds for ice number concentration with values of 6.66 cm-3 ... . For clean cirrus, typical values of concentration ... are close to 0.1 cm-3 ... (..Voigt et al., 2017). '

The authors compare ice number concentrations in different size ranges! 6.66cm-3 is for sizes larger than 0.4 micron, while 0.1 cm-3 is for sizes larger than 3 micron. Here, the value in parenthesis in Table 3 needs to be compared, which is 0.36/0.214 cm-3 (mean/median), which is in better agreement. The authors mention that later in the text, so the first statement (significantly higher values) needs to be removed.

**Answer from Authors:**

We thank the reviewer for these remarks and hope this last part of the paper becomes clearer.

Cirrus properties are generally analysed for ice particle diameters higher than 3 µm. Then, as recommended by the reviewer, only properties from this part of the PSD are analysed for clarity reason.

Indeed, IWC values of 0.001 mg m$^{-3}$ are not often reached (Voigt et al., 2017). Then, mean IWC values of 0.055 mg m$^{-3}$ have been retrieved for natural cirrus by Schumann et al. (2017) work. This value still significantly lower than observations from CONCERT campaigns due to a largely influenced atmosphere by air traffic. However, some studies, reported through Heymsfield et al. (2016), show consistent median IWC values with CONCERT campaigns for given temperatures.

**Modifications:**

l. 646: "*For aged contrails, concentrations of ice particles with sizes greater than 3 µm are below 0.5 cm$^{-3}$, which agree with concentrations presented in other contrail studies (close to 1 cm$^{-3}$ in Lawson et al., 1998 and Schumann et al., 2017). Also aged contrails IWC and extinction coefficients mean values lie between 0.3 mg m$^{-3}$ and 5.7 mg m$^{-3}$ and 0.09 and 0.2 km$^{-1}$, respectively. These values are only weakly sensitive to the used cut-off size (below or above 3 µm) and are in accordance with previous studies where IWC values up to 10 mg m$^{-3}$ and extinction coefficient below 0.5 km$^{-1}$ were measured (Schröder et al., 2000; Febvre et al., 2009; De Leon et al., 2012).*"

l. 654: "*Our results also show that differences greater than 5 m mg$^{-3}$ can be found within young contrails (YC1 and YC2) and aged contrails (AC1 and AC2) clusters. This variability could be attributed to a small number of large particles with diameter higher than 20 µm in YC1 and AC2 compared to YC2 and AC1 clusters.*"

[revised manuscript text omitted]